# Linearization Explains Fine-Tuning in Large Language Models

**Zahra Rahimi Afzal**[*][a]    **Tara Esmaeilbeig**[*][a,b]    **Mojtaba Soltanalian**[a]    **Mesrob I. Ohannessian**[a]

[a]University of Illinois Chicago    [b]Nokia Bell Labs
[a]{zrahim2, msol, mesrob}@uic.edu
[b]tara.esmaeilbeig@nokia.com

## Abstract

Parameter-Efficient Fine-Tuning (PEFT) is a popular class of techniques that strive to adapt large models in a scalable and resource-efficient manner. Yet, the mechanisms underlying their training performance and generalization remain underexplored. In this paper, we provide several insights into such fine-tuning through the lens of linearization. Fine-tuned models are often implicitly encouraged to remain close to the pretrained model. By making this explicit, using an $\ell_2$-distance inductive bias in parameter space, we show that fine-tuning dynamics become equivalent to learning with the positive-definite neural tangent kernel (NTK). We specifically analyze how close the fully linear and the linearized fine-tuning optimizations are, based on the strength of the regularization. This allows us to be pragmatic about how good a model linearization is when fine-tuning large language models (LLMs). When linearization is a good model, our findings reveal a strong correlation between the eigenvalue spectrum of the NTK and the performance of model adaptation. Motivated by this, we give spectral perturbation bounds on the NTK induced by the choice of layers selected for fine-tuning. We empirically validate our theory on Low Rank Adaptation (LoRA) on LLMs. These insights not only characterize fine-tuning but also have the potential to enhance PEFT techniques, paving the way to better informed and more nimble adaptation in LLMs.

## 1  Introduction

Foundational large language models (LLMs) [1, 2] have emerged as general-purpose models that are then adapted to various NLP tasks through fine-tuning. Due to the enormous number of trainable parameters in these models, full fine-tuning can be expensive in terms of time and other computational resources [3]. Parameter-Efficient Fine-tuning (PEFT) [4–8] is a particularly popular set of techniques that strive to tackle this computational burden. They aim to reduce the effective number of trained parameters, making choices such as focusing on certain layers only or applying rank-limited updates, all while maintaining how well the model adapts to the task. However, these methods often lack a fundamental understanding of the dynamics behind these choices, making informed exploration of the algorithmic space difficult.

In this paper, we introduce *linearized* fine-tuning, which establishes a foundation for rigorously understanding adaptation in large models through the lens of Neural Tangent Kernel (NTK) regression. Said simply, linearizing the fine-tuning process reduces the problem to one closely aligned with NTK regression. This, in turn, allows us to predict the performance of various fine-tuning decisions using the properties of the NTK kernel. Linearization can be achieved by regularizing fine-tuning to remain

---

[*]Equal contribution.

[†]Our code is publicly available at `https://github.com/zahrahimi/linearization`.

39th Conference on Neural Information Processing Systems (NeurIPS 2025).

close to the original model, which is empirically found to be non-restrictive, because most methods already encourage this proximity between the fine-tuned and original models [9].

Linearized approximations to guide fine-tuning have previously been explored in [10–14]. In [10], a linearized approximation is used to introduce two scores for selecting models from a zoo of models to be fine-tuned for a specific task. In [11], a linear model is constructed that combines network activation with the gradient features. In [12], it is demonstrated that linearized models, governed by the NTK, outperform their nonlinear counterparts in fine-tuning. All these works, however, do not quantify the closeness of fine-tuning dynamics to the linear approximation. Instead, they only hypothesize that the model will remain close to the pretrained model and that, therefore, the fine-tuning dynamics can be approximated by a Taylor expansion of the model around the pretrained parameters. This shortcoming motivates the present paper to introduce an explicit inductive bias that enables quantifying the extent to which linearity is preserved during fine-tuning. In particular, we provide a theoretical upper bound on the distance between the fine-tuned model and its linearized approximation, which in turn firmly supports predictions of fine-tuning performance based on linearization. The work in [13] explores the loss landscape in the NTK regime; however, it does not determine under what conditions fine-tuning falls within this regime. In contrast, our work proves both theoretically and empirically that, although necessarily in the NTK (linearization) regime, fine-tuning can be constrained to it by regularizing the process towards the pretrained model.

Our work builds on prior results that establish conditions under which linearity holds. Jacot et al. [15] showed that in the infinite width limit, the network function follows a linear differential equation during training. Moreover, they proved that for wide networks, the NTK remains constant during training; hence, they called this regime *lazy* training. Later, Bach et al. [16] suggested that lazy training is not limited to infinite-width networks.

More recently, and of particular relevance here, Malladi et al. in [17] extended the NTK theory to characterize kernel-based dynamics specifically for fine-tuning language models. They adapted infinite-width analysis to account for pretrained initialization and proved that prompt-based fine-tuning in complex architectures like transformers can exhibit kernel behavior under certain conditions. They demonstrated the critical role of meaningful prompts in achieving kernel behavior and explained how these dynamics describe fine-tuning trajectories. Additionally, they proved the effectiveness of Low Rank Adaptation (LoRA) by bounding the difference between the kernel of full fine-tuning and the kernel of LoRA. In [17], the authors assume that fine-tuning can be explained by linearization, and their results also confirm that linearization does not always appear in fine-tuning (see Fig. 1 of [17]). In our work, however, by introducing an explicit inductive bias for fine-tuning, which is essentially a weight decay toward the original model parameter, we induce lazy training. Consequently, the fine-tuning performance can be approximated by the linearized model [9, 18]. The evolution of the model is described by the neural tangent kernel (NTK), which stays constant during fine-tuning. This property makes it possible to study the generalization properties of a fine-tuned model based on the spectral properties of the NTK. Specifically, our main contributions are:

- We show that having an explicit inductive bias toward the pretrained model results in linearized fine-tuning; more precisely, it brings fine-tuned and linearized models closer.

- Leveraging the linearization of fine-tuning, we formulate the fine-tuning problem as a neural tangent kernel regression and use the eigenvalue decomposition of the neural tangent kernel to derive bounds on the empirical risk of the end result of fine-tuning, prior to its execution.

- We provide bounds on the spectral perturbation on the NTK when a set of trainable parameters is added to fine-tuning. In particular, to the best of our knowledge, we present the first spectrum perturbation results due to layer selection and introduce an algorithm that provides new insights to guide fine-tuning design.

- Through extensive experiments, we validate our theoretical results. We evaluate the condition number of NTK as an at-initialization metric to anticipate the performance of LoRA before training. Even though our experiments focus on LoRA, the technical tools we introduce could be equally used in the context of other PEFT methods.

The paper is organized as follows. In §2, we present the problem formulation and notation. In §3, we detail the linearization framework. In §4, we give an NTK regression formulation for the linearized fine-tuning and related performance to the kernel's spectrum. In §5, we show how layer choices affect this spectrum. We then validate our findings through experiments in §6 and conclude in §7.

## 2  Problem Formulation

In the fine-tuning problem, we are given a pretrained model $f_{\boldsymbol{\theta}_0}(\cdot)$, a target task dataset $\mathcal{D}_T = (\mathbf{x}_i, \mathbf{Y}_i)_{i=1}^n$ for the downstream task, and a loss function $\mathcal{L}(\cdot, \cdot) : \mathbb{R} \times \mathbb{R} \to \mathbb{R}$. The most basic fine-tuning solution is to minimize the target empirical risk via gradient descent, when initialized at $f_{\boldsymbol{\theta}_0}(\cdot)$. This implicitly finds solutions close to the pretrained model. In this paper, we make this explicit via an inductive bias toward the pretrained model. We denote the regularized fine-tuned model by $f_{\boldsymbol{\theta}^\star}(\cdot) : \mathbb{R}^d \to \mathbb{R}$, and we define it as minimizing the regularized empirical risk.

$$\boldsymbol{\theta}^\star = \underset{\boldsymbol{\theta}}{\text{minimize}}\ \widetilde{\mathcal{R}}(\boldsymbol{\theta}) + \frac{\lambda}{2}\|\boldsymbol{\theta} - \boldsymbol{\theta}_0\|_2^2, \quad \text{where} \tag{1}$$

$$\widetilde{\mathcal{R}}(\boldsymbol{\theta}) = \sum_{i=1}^n \mathcal{L}(f_{\boldsymbol{\theta}}(\mathbf{x}_i), \mathbf{y}_i). \tag{2}$$

$\boldsymbol{\theta}$ denotes the trainable fine-tuning parameters, $\boldsymbol{\theta}_0$ denotes the parameters of the pretrained model, $\widetilde{\mathcal{R}}(\boldsymbol{\theta})$ the original objective function, and $\lambda$ the regularization strength hyperparameter. When $\boldsymbol{\theta}_0 = 0$, (1) reduces to an ordinary weight decay. This regularization reduces the deviation between the fine-tuned and pretrained models, and is a simple instance of proximal methods often used in fine-tuning. We assume $\mathcal{L}(\cdot, \cdot)$ is the squared loss, and thus the risk is the MSE. By representing $\mathcal{D}_T = (\mathbf{x}_i, \mathbf{y}_i)_{i=1}^n$ as $\mathbf{X} = [\mathbf{x}_1, \mathbf{x}_2, \ldots, \mathbf{x}_n]^\top \in \mathbb{R}^{n \times d}$, and $\mathbf{Y} = [\mathbf{y}_1, \mathbf{y}_2, \ldots, \mathbf{y}_n] \in \mathbb{R}^{1 \times n}$, we write the risk as

$$\widetilde{\mathcal{R}}(\boldsymbol{\theta}) = \|f_{\boldsymbol{\theta}}(\mathbf{X}) - \mathbf{Y}\|^2, \tag{3}$$

where $f_{\boldsymbol{\theta}}(\mathbf{X}) = [f_{\boldsymbol{\theta}}(\mathbf{x}_1), \ldots, f_{\boldsymbol{\theta}}(\mathbf{x}_n)] \in \mathbb{R}^{1 \times n}$. We split the optimization (1) using gradient descent at step $t$ with learning rate $\eta$ into two steps:

$$\widetilde{\boldsymbol{\theta}}_t = \boldsymbol{\theta}_{t-1} - \eta \nabla \widetilde{\mathcal{R}}(\boldsymbol{\theta}_{t-1})^\top, \tag{4}$$

$$\boldsymbol{\theta}_t = \widetilde{\boldsymbol{\theta}}_t - \lambda \eta \left(\widetilde{\boldsymbol{\theta}}_t - \boldsymbol{\theta}_0\right), \tag{5}$$

where the first step is the gradient descent on $\widetilde{\mathcal{R}}(\boldsymbol{\theta})$ and the second step is the gradient descent on the regularization term $\frac{\lambda}{2}\|\boldsymbol{\theta} - \boldsymbol{\theta}_0\|^2$.

## 3  Proximity to the Pretrained Model Promotes Linearity

We first justify why we regularize for proximity to the pretrained model by showing that this promotes similarity between the final fine-tuned solution and its linearized counterpart. For any $(\mathbf{x}, \mathbf{y}) \in \mathcal{D}_T$, the latter is defined as

$$\bar{f}_{\bar{\boldsymbol{\theta}}_t}(\mathbf{x}) = f_{\boldsymbol{\theta}_0}(\mathbf{x}) + \left\langle \nabla f_{\boldsymbol{\theta}_0}(\mathbf{x}), \bar{\boldsymbol{\theta}}_t - \boldsymbol{\theta}_0 \right\rangle, \tag{6}$$

where $\bar{\boldsymbol{\theta}}$ denotes the parameters of the linearized model. We set $\bar{\boldsymbol{\theta}}_0 = \boldsymbol{\theta}_0$ and, therefore, for any $\mathbf{x} \in \mathcal{D}_T$, $\bar{f}_{\bar{\boldsymbol{\theta}}_0}(\mathbf{x}) = f_{\boldsymbol{\theta}_0}(\mathbf{x})$.

We list and discuss our main theorems toward this goal, and defer the proofs to the appendix. In Theorem 1, we provide conditions for monotonicity of updates in regularized fine-tuning, and use it to slightly modify gradient descent to ensure monotonicity. In Theorem 2, we derive an upper bound on the distance between the *parameters* at a step $t$ of the regularized fine-tuned model and the pretrained model, i.e., $\|\boldsymbol{\theta}_t - \boldsymbol{\theta}_0\|$. In Theorem 3, we find the distance

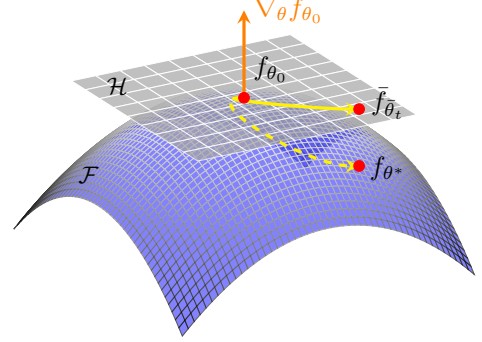

Figure 1: The NTK defines a linear function space $\mathcal{H}$ tangent to the non-linear function space $\mathcal{F}$ defined by the model. Regularized fine-tuning in the lazy regime is close to kernel regression on the tangent space. $f_{\boldsymbol{\theta}^\star}(\mathbf{x})$ is the fine-tuned model obtained by empirical risk minimization. If fine-tuning remains in the linearized regime, then after $T$ steps of training $f_{\boldsymbol{\theta}^\star}(\mathbf{x}) \approx f_{\boldsymbol{\theta}_0}(\mathbf{x}) + \left\langle \nabla_{\boldsymbol{\theta}} f_{\boldsymbol{\theta}_0}(\mathbf{x}), \bar{\boldsymbol{\theta}}_T - \boldsymbol{\theta}_0 \right\rangle$ is a good approximation.

between the *inference* of the regularized fine-tuned model $f_{\boldsymbol{\theta}}(\mathbf{X})$ and the linearized model $\bar{f}_{\bar{\boldsymbol{\theta}}}(\mathbf{X})$ i.e., $\|f_{\boldsymbol{\theta}}(\mathbf{X}) - \bar{f}_{\bar{\boldsymbol{\theta}}}(\mathbf{X})\|$, which is the primary contribution of this section.

**Theorem 1.** *Under the squared loss, for any $t > 0$, if $\lambda > 0$ and $\nabla_{\boldsymbol{\theta}} \widetilde{\mathcal{R}}(\boldsymbol{\theta}_t)(\boldsymbol{\theta}_t - \boldsymbol{\theta}_0) \geq 0$, then*

$$\frac{d}{dt}\|f_{\theta_t}(\mathbf{X}) - \mathbf{Y}\|^2 \leq 0. \tag{7}$$

*Moreover, if $\nabla_{\boldsymbol{\theta}} \widetilde{\mathcal{R}}(\boldsymbol{\theta}_t)(\boldsymbol{\theta}_t - \boldsymbol{\theta}_0) < 0$, then $\lambda = 0$ is a sufficient condition for (7) to hold.*

*Proof.* See Appendix B. □

In fine-tuning, the objective in (1) is non-convex in the parameters. However, according to Theorem 1, a sufficient condition for regularized fine-tuning to have non-increasing $\|f_{\theta_t}(\mathbf{X}) - \mathbf{Y}\|_2^2$, is that at step $t$, we have $\nabla_{\boldsymbol{\theta}_t} \widetilde{\mathcal{R}}(\boldsymbol{\theta}_t)(\boldsymbol{\theta}_t - \boldsymbol{\theta}_0) \geq 0$ or $\lambda = 0$. Therefore, we modify the regularization step in (5) as

$$\boldsymbol{\theta}_t = \begin{cases} \widetilde{\boldsymbol{\theta}}_t - \lambda\eta\left(\widetilde{\boldsymbol{\theta}}_t - \boldsymbol{\theta}_0\right) & \text{if } \nabla_{\boldsymbol{\theta}_t} \widetilde{\mathcal{R}}(\boldsymbol{\theta}_t)(\boldsymbol{\theta}_t - \boldsymbol{\theta}_0) \geq 0, \\ \widetilde{\boldsymbol{\theta}}_t & \text{if } \nabla_{\boldsymbol{\theta}_t} \widetilde{\mathcal{R}}(\boldsymbol{\theta}_t)(\boldsymbol{\theta}_t - \boldsymbol{\theta}_0) < 0 \end{cases}. \tag{8}$$

This is a selective regularization scheme, which decides on regularization of each trainable parameter based on the direction of its gradient. Intuitively, the inner product $\nabla_{\boldsymbol{\theta}_t} \widetilde{\mathcal{R}}(\boldsymbol{\theta}_t)(\boldsymbol{\theta}_t - \boldsymbol{\theta}_0)$ reflects how aligned the current progress is with the negative gradient update direction. A negative inner product signals consistent improvement in loss, while a positive value suggests movement toward a higher-loss region. As a result, in order to have non-increasing loss at step $t$, (8) regularizes only the trainable parameters with positive inner product values [9].

**Theorem 2.** *Consider the selectively regularized fine-tuning solution, under squared loss. Denote the instantaneous value of the regularization parameter by $\lambda_t$, which can be either $0$ or $\lambda$. If $f_{\boldsymbol{\theta}}(\mathbf{X})$ is $Lip(f)$-Lipschitz in an $\ell_2$-ball of radius $r$ around pretrained parameters $\boldsymbol{\theta}_0$, we have*

$$\|\boldsymbol{\theta}_t - \boldsymbol{\theta}_0\| \leq 2\,Lip(f)\,\|f_{\boldsymbol{\theta}_0}(\mathbf{X}) - \mathbf{Y}\|\int_0^t e^{-(\Lambda_t - \Lambda_s)}\,ds, \quad \text{where } \Lambda_t = \int_0^t \lambda_s ds. \tag{9}$$

*In the special case when $\lambda_t = \lambda$ (remains constant), we obtain*

$$\|\boldsymbol{\theta}_t - \boldsymbol{\theta}_0\| \leq 2\,Lip(f)\|f_{\boldsymbol{\theta}_0}(\mathbf{X}) - \mathbf{Y}\|\frac{1 - e^{-\lambda t}}{\lambda}. \tag{10}$$

*Proof.* See Appendix C. □

Intuitively, one can think of (9) as placing the "average" $\lambda$, $\frac{1}{t}\Lambda_t$ in (10). In what follows, we adhere to (10), for simplicity. Note that in conventional fine-tuning, $\lambda = 0$. Since $\lim_{\lambda \to 0} \frac{1 - e^{-\lambda t}}{\lambda} = t$ the bound in (10) recovers

$$\|\boldsymbol{\theta}_t - \boldsymbol{\theta}_0\| \leq 2\,Lip(f)\,\|f_{\boldsymbol{\theta}_0}(\mathbf{X}) - \mathbf{Y}\|\,t, \tag{11}$$

which follows Theorem 2.3 in [16], in the case when the rescaling parameter is one. In other words, regularization and rescaling both manifest the lazy training phenomenon, but regularization maintains much closer proximity, since $\frac{1 - e^{-\lambda t}}{\lambda}$ is bounded, unlike $t$.

Theorem 2 is primarily a building block for Theorem 3; it simply says that while the solution may deviate from the origin under regularization, we can bound that deviation. Theorems 3 and 4 show that we can jump from a deviation from the initialization bound, which is the direct objective of regularization, to a bound on how far the NTK solution and the regularized fine-tuning solutions are, i.e., the indirect but main benefit of regularization. This is what allows us to place the approximation of Section §4 on a stronger footing than prior works [10–13] that assume that such proximity simply holds. For simplicity, we consider the $\lambda$ constant case to prove Theorems 3 and 4.

**Theorem 3.** *Under the squared loss, if $f_{\boldsymbol{\theta}}(\mathbf{X})$ and $\nabla f_{\boldsymbol{\theta}}(\mathbf{X})$ are $Lip(f)$-Lipschitz and $Lip(\nabla f)$-Lipschitz in an $\ell_2$-ball of radius $r$ around $\boldsymbol{\theta}_0$ respectively, we have*

$$\|f_{\boldsymbol{\theta}_t}(\mathbf{X}) - \bar{f}_{\bar{\boldsymbol{\theta}}_t}(\mathbf{X})\| \leq b\left(t - \frac{1 - e^{-\lambda t}}{\lambda}\right), \quad \text{where} \tag{12}$$

$$b = 2\,Lip(f)^2\|f_{\boldsymbol{\theta}_0}(\mathbf{X}) - \mathbf{Y}\|\left(\frac{4}{\lambda}Lip(\nabla f)\|f_{\boldsymbol{\theta}_0}(\mathbf{X}) - \mathbf{Y}\| + 1\right). \tag{13}$$

*Proof.* See Appendix D. □

In the following theorem, we show that, for a proper choice of the regularization parameter $\lambda$, linearization of fine-tuning only depends on the local properties of $f_{\boldsymbol{\theta}}(\mathbf{X})$ around $\boldsymbol{\theta}_0$.

**Theorem 4.** *Let the loss be the squared loss, $f_{\boldsymbol{\theta}}(\mathbf{X})$ and $\nabla f_{\boldsymbol{\theta}}(\mathbf{X})$ be Lip(f)-Lipschitz and Lip($\nabla f$)-Lipschitz, respectively, in an $\ell_2$-ball of radius $r$ around $\boldsymbol{\theta}_0$. Define*

$$\lambda_\circ = \frac{2\|f_{\boldsymbol{\theta}_0}(\mathbf{X}) - \mathbf{Y}\|\, Lip(f)}{r}.$$

- *If $\lambda \geq \lambda_\circ$, then for all $t$, the following holds.*

- *If $\lambda < \lambda_\circ$, then the following holds for $t \leq \frac{1}{\lambda} \ln \frac{1}{1 - \lambda/\lambda_\circ}$*

$$\|f_{\boldsymbol{\theta}_t}(\mathbf{X}) - \bar{f}_{\bar{\boldsymbol{\theta}}_t}(\mathbf{X})\| \leq 2\, Lip(f)^2 \|\mathbf{Y}(0) - \mathbf{Y}\| \left( \frac{4}{\lambda} Lip(\nabla f)\|\mathbf{Y}(0) - \mathbf{Y}\| + 1 \right) t.$$

*In particular, for $\lambda \geq \lambda_\circ$, the bound from Theorem 3 always holds and simplifies to:*

$$\|f_{\boldsymbol{\theta}_t}(\mathbf{X}) - \bar{f}_{\bar{\boldsymbol{\theta}}_t}(\mathbf{X})\| \leq 2\, Lip(f)\widetilde{R}(\boldsymbol{\theta}_0)\, (2r Lip(\nabla f) + Lip(f))\, t.$$

*Proof.* See Appendix E. □

While the distance between the pretrained and fine-tuned models can be substantial, the results illustrate that this gap can be effectively constrained through regularization. Under this condition, fine-tuning can be modeled by NTK regression [15], as discussed in the following section.

The takeaway of Theorem 4 is that the regularization determines the time scale at which the approximation holds, as well as how tight the approximation is. Theorem 4, at first glance, seems to suggest that a larger value of $\lambda$ is better to bring NTK and regularized trajectories together. However, a larger $\lambda$ means that we effectively force ourselves to remain in an even smaller ball around $\boldsymbol{\theta}_0$ and may suffer in terms of fine-tuning performance (see $\lambda = 50$ in Table 1). The "sweet spot" regularization parameter can be read from Theorem 4. It should be proportional to the smoothness of $f$, the pretrained error, and inversely proportional to the radius where the smoothness assumptions hold. While $\lambda_\circ$ is in terms of Lip($f$), Theorem 3 and (13) show that there are also diminishing returns for pushing $\lambda$ beyond Lip($\nabla f$), due to the $+1$ in this term.

## 4  Fine-Tuning Meets Neural Tangent Kernel Regression

We formally define the fine-tuning problem as a regularized function estimation in the reproducing kernel Hilbert space (RKHS), $\mathcal{H}$, generated by the NTK, $\mathbf{k}(\mathbf{x}, \mathbf{x}') = \nabla f_{\boldsymbol{\theta}}(\mathbf{x})\nabla f_{\boldsymbol{\theta}}(\mathbf{x}')^\top$.

As shown in the previous section, regularized fine-tuning is in the linearized regime. Therefore, $f_{\boldsymbol{\theta}^\star}(\cdot)$ can be approximated by its dual in the tangent space, as shown in Figure 1. We can interpret the linearized model as a kernel method with a feature map given by $\phi(\mathbf{x}) = \nabla_{\boldsymbol{\theta}} f_{\boldsymbol{\theta}_0}(\mathbf{x})$. The corresponding kernel induced by this feature map is the NTK. Jacot et al. in [15], showed that gradient descent with an infinitesimally small learning rate is equivalent to performing kernel regression with the fixed NTK. Thus, the empirical risk minimization (1) is approximated by kernel regression when the kernel is the NTK [15].

Despite the fact that NTK analysis often considers the entire model, it is straightforward to specialize it to the fine-tuning case, which we do here for completeness. When using mean squared error, fine-tuning with the linear model is similarly equivalent to solving the kernel regression problem presented in (14). Let $\mathcal{H}$ be the reproducing kernel Hilbert space (RKHS) endowed with a positive definite kernel function $\mathbf{k}(\cdot, \cdot)$, i.e.,

$$\mathcal{H} = \left\{ f(\cdot) = \sum_{i=1}^n \alpha_i \mathbf{k}(\cdot, \mathbf{x}_i) \right\}.$$

Assuming the solution lies in or close to this Hilbert space, then as an alternative to (1), we solve

$$\underset{f \in \mathcal{H}}{\text{minimize}} \quad \frac{1}{n} \sum_{(\mathbf{x},\mathbf{y}) \in \mathcal{D}_T} \|f(\mathbf{x}) - \mathbf{y}\|^2 + \sigma \|f\|_{\mathcal{H}}^2, \tag{14}$$

where $\| \cdot \|_{\mathcal{H}}$ is the norm corresponding to the inner product $\langle \cdot, \cdot \rangle_{\mathcal{H}}$ defined on the RKHS $\mathcal{H}$. The parameter $\sigma > 0$ regularizes the problem toward minimum-norm solutions and is qualitatively inversely related to the number of gradient descent steps in the original problem. We thus effectively formulate the fine-tuning problem as a regularized function estimation in the RKHS, $\mathcal{H}$, generated by the NTK, $\mathbf{k}(\mathbf{x}, \mathbf{x}') = \nabla f_{\boldsymbol{\theta}_0}(\mathbf{x}) \nabla f_{\boldsymbol{\theta}_0}(\mathbf{x}')^{\top}$. According to the representer theorem [19], the problem (14) on training dataset $\mathcal{D}_T = \{\mathbf{x}_i, \mathbf{y}_i\}_{i=1}^n$ possesses the closed-form solution

$$f^*(\cdot) = \sum_{i=1}^n \alpha_i \mathbf{k}(\cdot, \mathbf{x}_i) = \boldsymbol{\alpha}^{\top} \mathbf{K}(\cdot, \mathbf{X}), \tag{15}$$

where $\mathbf{K}(\cdot, \mathbf{X}) = [\mathbf{k}(\cdot, \mathbf{x}_1), \ldots, \mathbf{k}(\cdot, \mathbf{x}_n)] \in \mathbb{R}^{1 \times n}$. Substituting (15) in (14), we have

$$\underset{\boldsymbol{\alpha}}{\text{minimize}} \quad \mathbb{E}_{(\mathbf{x},\mathbf{y}) \sim \mathcal{D}_T} \left\| \boldsymbol{\alpha}^{\top} \mathbf{K}(\cdot, \mathbf{X}) - \mathbf{Y} \right\|^2 + \sigma \|f\|_{\mathcal{H}}^2, \tag{16}$$

which is a convex problem with $\boldsymbol{\alpha}^* = [\mathbf{K}(\mathbf{X}, \mathbf{X}) + \sigma \mathbf{I}]^{-1} \mathbf{Y}$ as the solution. Equivalently,

$$f^*(\cdot) = \mathbf{K}(\cdot, \mathbf{X}) [\mathbf{K}(\mathbf{X}, \mathbf{X}) + \sigma \mathbf{I}]^{-1} \mathbf{Y}, \tag{17}$$

where $[\mathbf{K}(\mathbf{X}, \mathbf{X})]_{i,j} = \mathbf{k}(\mathbf{x}_i, \mathbf{x}_j)$. For the sake of brevity, hereafter, we use $\mathbf{K}(\mathbf{X}, \mathbf{X})$ and $\mathbf{K}$ interchangeably.

Under linearity, we now show through Theorem 5 that the spectrum of the NTK directly affects the empirical risk, making it a tool that can be used for making fine-tuning decisions.

**Theorem 5.** *The empirical risk is bounded as*

$$\left( \frac{\sigma \|\mathbf{Y}\|}{\sigma + \lambda_{max}(\mathbf{K})} \right)^2 \leq \mathcal{R}(\boldsymbol{\theta}) \leq \left( \frac{\sigma \|\mathbf{Y}\|}{\sigma + \lambda_{min}(\mathbf{K})} \right)^2, \tag{18}$$

*where $\lambda_{min}(\mathbf{K})$ and $\lambda_{max}(\mathbf{K})$ are the minimum and maximum eigenvalues of $\mathbf{K}(\mathbf{X}, \mathbf{X})$, respectively.*

*Proof.* See Appendix F. $\qquad\square$

Theorem 5 motivates us to study the regularized condition number $\kappa(\mathbf{K} + \sigma \mathbf{I}) = \frac{\lambda_{\max}(\mathbf{K}) + \sigma}{\lambda_{\min}(\mathbf{K}) + \sigma}$ as an at-initialization metric for predicting the performance of fine-tuning. To show how this can be useful, for example, in selecting what subset of parameters to tune, we next study the randomized kernel specifically for fine-tuning, which enables us to investigate the training dynamics based on the influence of each individual trainable parameter on the bounds in Theorem 5.

## 5 Spectral Perturbation of Layers

We have shown that (i) regularized fine-tuning is well-approximated by its fully linearized counterpart, (ii) that optimizing the linearized fine-tuning is equivalent to NTK kernel regression, and (iii) that the kernel's spectrum is predictive of the fine-tuning performance. To put these facts to use, in this section, we study the effect of layer selection on the spectrum of the NTK, and consequently on the empirical risk of fine-tuning. Let $\boldsymbol{\theta}^l$ be the parameters of the layer $l$ from the pretrained model and $\boldsymbol{\theta} = \{\boldsymbol{\theta}^1, \ldots, \boldsymbol{\theta}^L\}$ be the set of parameters of the $L$ layers from the pretrained model. The NTK, when the parameters in $\boldsymbol{\theta}$ are chosen for fine-tuning, is $[\mathbf{K}]_{i,j} = \sum_{l=1}^L \nabla_{\boldsymbol{\theta}^l} f_{\boldsymbol{\theta}}(\mathbf{x}_i) \nabla_{\boldsymbol{\theta}^l} f_{\boldsymbol{\theta}}(\mathbf{x}_j)^{\top}$. The kernel induced by $\boldsymbol{\theta}^l$ is $\mathbf{S}_l \in \mathbb{R}^{n \times n}$, where

$$[\mathbf{S}_l]_{i,j} = \nabla_{\boldsymbol{\theta}^l} f_{\boldsymbol{\theta}}(\mathbf{x}_i) \nabla_{\boldsymbol{\theta}^l} f_{\boldsymbol{\theta}}(\mathbf{x}_j)^{\top}. \tag{19}$$

One can infer that the NTK induced by all layers is $\mathbf{K} = \sum_{l=1}^L \mathbf{S}_l$.

The following theorems quantify the effect of adding a set of trainable parameters on the spectrum of the NTK.

| Dataset | Hyper-Parameter $\lambda$ | 50 | 10 | 5 | 2 | 1 | 0.5 | 0.1 | 0.0 |
|---------|---------------------------|------|------|------|------|------|------|------|------|
| **CoLA** | $\|\boldsymbol{\theta}_t - \boldsymbol{\theta}_0\|_2$ | 0.280 | 0.350 | 0.404 | 0.5263 | 0.6148 | 0.6946 | 0.8223 | 0.960 |
| | $\|f_{\boldsymbol{\theta}_t}(\mathbf{X}) - \bar{f}_{\bar{\boldsymbol{\theta}}_t}(\mathbf{X})\|_2$ | 1.06 | 1.12 | 1.39 | 1.25 | 1.27 | 1.32 | 1.28 | 1.47 |
| | **KL Divergence** | 0.1060 | 0.1377 | 0.200 | 0.1613 | 0.1788 | 0.1961 | 0.1599 | 0.210 |
| | **Evaluation Accuracy of** $f_{\boldsymbol{\theta}_t}(\mathbf{x})$ | 74.59 | 79.57 | 80.44 | 79.38 | 80.24 | 80.15 | 80.15 | 79.67 |
| **SST-2** | $\|\boldsymbol{\theta}_t - \boldsymbol{\theta}_0\|_2$ | 0.292 | 0.336 | 0.369 | 0.424 | 0.520 | 0.700 | 1.589 | 2.519 |
| | $\|f_{\boldsymbol{\theta}_t}(\mathbf{X}) - \bar{f}_{\bar{\boldsymbol{\theta}}_t}(\mathbf{X})\|_2$ | 1.712 | 2.303 | 2.635 | 2.957 | 3.217 | 3.331 | 3.397 | 2.791 |
| | **KL Divergence** | 0.320 | 0.433 | 0.476 | 0.517 | 0.545 | 0.560 | 0.578 | 0.540 |
| | **Evaluation Accuracy of** $f_{\boldsymbol{\theta}_t}(\mathbf{x})$ | 89.3 | 91.2 | 91.5 | 92.4 | 92.8 | 93.0 | 92.4 | 91.6 |

Table 1: Sweep over the hyperparameter ($\lambda$). Increasing regularization strength, i.e., larger $\lambda$, reduces the deviation between the regularized fine-tuning and linearized models at one snapshot of fine-tuning at step $t$. Accuracy is largely unaffected by regularization.

**Theorem 6.** *Let $\mathbf{K}$ be the NTK with respect to the set of selected fine-tuning parameters, and $\mathbf{S}$ be the kernel with respect to the parameters of the candidate layers, to add to the fine-tuning parameters. Then*

$$(1 - \eta)\lambda_i(\mathbf{K}) \leq \lambda_i(\mathbf{K} + \mathbf{S}) \leq (1 + \eta)\lambda_i(\mathbf{K}), \tag{20}$$

*where $\eta = \|\mathbf{K}^{-1/2}\mathbf{S}\,\mathbf{K}^{-1/2}\|$.*

*Proof.* See Appendix G. $\qquad\square$

The spectral perturbation bounds of Theorem 6 can interact with the risk bounds given by Theorem 5, to help us further understand the relative merits of parameter choices. We denote by $\mathcal{R}(\boldsymbol{\theta})$ and $\mathcal{R}(\boldsymbol{\theta} \cup \hat{\boldsymbol{\theta}})$, respectively, the empirical risk of the model fine-tuned with parameters $\boldsymbol{\theta}$ and $\boldsymbol{\theta} \cup \hat{\boldsymbol{\theta}}$.

**Theorem 7.** *Let $\mathbf{K}$ be the NTK induced by the trainable parameters in $\boldsymbol{\theta}$, then if $\kappa(\mathbf{K} + \sigma\mathbf{I}) \leq c$, we have*

$$\frac{\lambda_{max}(\mathbf{K} + \mathbf{S} + \sigma\mathbf{I})}{a\lambda_{max}(\mathbf{K} + \sigma\mathbf{I})} \leq \left(\frac{\mathcal{R}(\boldsymbol{\theta} \cup \hat{\boldsymbol{\theta}})}{\mathcal{R}(\boldsymbol{\theta})}\right)^{\frac{1}{2}} \leq \frac{a\lambda_{max}(\mathbf{K} + \mathbf{S} + \sigma\mathbf{I})}{\lambda_{max}(\mathbf{K} + \sigma\mathbf{I})}, \tag{21}$$

*where $a = \frac{c}{(1-\eta)^2}$, $\eta = \|\mathbf{K}^{-1/2}\mathbf{S}\mathbf{K}^{-1/2}\|$ and $\mathbf{S}$ is the kernel induced by $\hat{\boldsymbol{\theta}}$ with $[\mathbf{S}]_{i,j} = \nabla_{\hat{\boldsymbol{\theta}}}f_{\boldsymbol{\theta}}(\mathbf{x}_i)\nabla_{\hat{\boldsymbol{\theta}}}f_{\boldsymbol{\theta}}(\mathbf{x}_j)^\top$.*

*Proof.* See Appendix H. $\qquad\square$

The above result implies that, by incorporating an additional layer to the parameters being fine-tuned, one can expect $\frac{1}{2}\log\frac{\mathcal{R}(\boldsymbol{\theta}\cup\hat{\boldsymbol{\theta}})}{\mathcal{R}(\boldsymbol{\theta})} \in \left[\log\frac{\lambda_{\max}(\mathbf{K}+\mathbf{S})+\sigma}{\lambda_{\max}(\mathbf{K})+\sigma} - \log a, \log\frac{\lambda_{\max}(\mathbf{K}+\mathbf{S})+\sigma}{\lambda_{\max}(\mathbf{K})+\sigma} + \log a\right]$.

**Corollary 1.** *When we have two candidate layers $\hat{\boldsymbol{\theta}}^1$ and $\hat{\boldsymbol{\theta}}^2$, mutatis mutandis, we have*

$$\frac{\lambda_{max}(\mathbf{K} + \mathbf{S}_1 + \sigma\mathbf{I})}{b\lambda_{max}(\mathbf{K} + \mathbf{S}_2 + \sigma\mathbf{I})} \leq \left(\frac{\mathcal{R}(\boldsymbol{\theta} \cup \hat{\boldsymbol{\theta}}^1)}{\mathcal{R}(\boldsymbol{\theta} \cup \hat{\boldsymbol{\theta}}^2)}\right)^{\frac{1}{2}} \leq \frac{b\lambda_{max}(\mathbf{K} + \mathbf{S}_1 + \sigma\mathbf{I})}{\lambda_{max}(\mathbf{K} + \mathbf{S}_2 + \sigma\mathbf{I})}, \tag{22}$$

*where $b = a_1 a_2$, $a_l = \frac{c}{(1-\eta_l)^2}$, $\eta_l = \|\mathbf{K}^{-1/2}\mathbf{S}_l\,\mathbf{K}^{-1/2}\|$ for $l \in \{1, 2\}$, and $\mathbf{S}_l$ is the kernel induced by $\hat{\boldsymbol{\theta}}^l$ with $[\mathbf{S}_l]_{i,j} = \nabla_{\hat{\boldsymbol{\theta}}^l}f_{\boldsymbol{\theta}}(\mathbf{x}_i)\nabla_{\hat{\boldsymbol{\theta}}^l}f_{\boldsymbol{\theta}}(\mathbf{x}_j)^\top$.*

As such, one can evaluate the candidacy of layers for fine-tuning based on Corollary 1. In order to anticipate the empirical risk of fine-tuning, at initialization we look at a confined interval, i.e., $\frac{1}{2}\log\frac{\mathcal{R}(\boldsymbol{\theta}\cup\hat{\boldsymbol{\theta}}^1)}{\mathcal{R}(\boldsymbol{\theta}\cup\hat{\boldsymbol{\theta}}^2)} \in \left[\log\frac{\lambda_{\max}(\mathbf{K}+\mathbf{S}_1)+\sigma}{\lambda_{\max}(\mathbf{K}+\mathbf{S}_2)+\sigma} - \log b, \log\frac{\lambda_{\max}(\mathbf{K}+\mathbf{S}_1)+\sigma}{\lambda_{\max}(\mathbf{K}+\mathbf{S}_2)+\sigma} + \log b\right]$.

## 6 Experiments

We now demonstrate that insights from our theory apply in practice, despite moving from squared loss to cross-entropy and from gradient descent to the AdamW optimizer, which we use for all fine-tuning in conjunction with the selective regularization (8). In our experiments, we implement LoRA on RoBERTa base and evaluate its performance on the General Language Understanding Evaluation (GLUE) benchmark [20], IMDb [21], and Yelp [22] datasets.

## 6.1 Model, Datasets and Optimizer

RoBERTa base incorporates several key modifications to the pretraining process, such as using larger batch sizes, longer sequences, and more diverse data than its antecedents like BERT [23]. Despite its relatively compact size of 125M parameters, RoBERTa base has proven to be one of the most powerful models for various NLP tasks, including text classification, question answering, and named entity recognition, especially on the GLUE benchmark.

The GLUE benchmark is a collection of diverse tasks that test a model's natural language understanding abilities. The tasks included in our experiments are linguistic acceptability judgment (CoLA [24]) and sentiment analysis (SST-2 [25]). The GLUE benchmark provides a comprehensive evaluation of a model's performance across various NLP challenges, assessing its ability to understand and reason about language in different contexts.

The IMDb dataset is a large dataset for binary sentiment classification, containing 50k highly popular movie reviews from the Internet Movie Database (IMDb). The Yelp dataset contains customer reviews from Yelp, a popular platform for crowd-sourced reviews about businesses, primarily restaurants. This dataset originally contains reviews with ratings from 1 to 5. To convert it into a binary classification task, we consider reviews with ratings less than 3 as label 0 (negative sentiment) and those with ratings greater than or equal to 3 as label 1 (positive sentiment).

Table 9 in Appendix N shows specific hyperparameters for RoBERTa base across various benchmarks, including GLUE tasks (CoLA, SST-2), Yelp, and IMDb. For all experiments, we use LoRA on the RoBERTa-base model from the Hugging Face transformers library [26], and report its performance on different tasks. We implemented our code on NVIDIA Tesla V100 GPUs. Following Hu et al. (2022), we mostly use the weights of the query and value layers, $\mathbf{W}_q \in \mathbb{R}^{m \times p}$ and $\mathbf{W}_v \in \mathbb{R}^{m \times p}$ for fine-tuning. In our experiments, we apply LoRA with $r = 8$, which has $(m + p) \times r = 2 \times 768 \times 8$ trainable parameters per selected layer for each of the query, key, and value projection matrices in the self-attention mechanism in the RoBERTa base model.

## 6.2 Linearization of Fine-Tuning

In Table 1, we analyze how regularization impacts the fine-tuning of models on the CoLA and SST-2 datasets, using several metrics, including the deviation of fine-tuned parameters from the pretrained parameters, $\|\boldsymbol{\theta}_t - \boldsymbol{\theta}_0\|$, KL divergence between predictions of the regularized fine-tuned model and its linearized counterpart, the $\ell_2$ norm $\|f_{\boldsymbol{\theta}_t}(\mathbf{X}) - \bar{f}_{\bar{\boldsymbol{\theta}}_t}(\mathbf{X})\|_2$, evaluation accuracy. The results indicate that increasing the regularization strength, $\lambda$, reduces both the deviation in parameters and prediction divergence, measured by KL divergence and $\|f_{\boldsymbol{\theta}_t}(\mathbf{X}) - \bar{f}_{\bar{\boldsymbol{\theta}}_t}(\mathbf{X})\|$, implying that stronger regularization keeps the fine-tuning behavior closer to its linearized approximation. Conversely, as regularization weakens, $\lambda \to 0$, parameter deviations and divergences from linearization increase, suggesting a more significant departure from the linear regime. This is well aligned with the bounds

| Dataset | Selected Layers | Selected Parameters | Condition Number | Train Loss | Evaluation Loss | Evaluation Accuracy |
|---------|-----------------|---------------------|------------------|------------|-----------------|---------------------|
| CoLA | {0} | $\{\mathbf{W}_q, \mathbf{W}_v\}$ | 11,618 | 0.5271 | 0.5265 | 73.15 |
| | {0,11} | $\{\mathbf{W}_q, \mathbf{W}_v\}$ | 9,490 | 0.5174 | 0.5293 | 73.15 |
| | {0,5,11} | $\{\mathbf{W}_q, \mathbf{W}_v\}$ | 7,503 | 0.5093 | 0.5272 | 73.44 |
| | {0,5,11} | $\{\mathbf{W}_k\}$ | 2,320 | 0.5128 | 0.5357 | 73.25 |
| SST-2 | {0} | $\{\mathbf{W}_q, \mathbf{W}_v\}$ | 5,195 | 0.4794 | 0.3871 | 83.48 |
| | {0,11} | $\{\mathbf{W}_q, \mathbf{W}_v\}$ | 6,413 | 0.4677 | 0.3916 | 82.79 |
| | {0,5,11} | $\{\mathbf{W}_q, \mathbf{W}_v\}$ | 6,792 | 0.5078 | 0.3913 | 82.68 |
| | {0,5,11} | $\{\mathbf{W}_k\}$ | 450.64 | 0.4717 | 0.3893 | 83.60 |
| Yelp | {0} | $\{\mathbf{W}_q, \mathbf{W}_v\}$ | 274 | 0.29 | 0.2597 | 88.20 |
| | {0,11} | $\{\mathbf{W}_q, \mathbf{W}_v\}$ | 4,167 | 0.2882 | 0.2596 | 88.24 |
| | {0, 5, 11} | $\{\mathbf{W}_q, \mathbf{W}_v\}$ | 1,336 | 0.2885 | 0.2596 | 88.21 |
| | {0,5,11} | $\{\mathbf{W}_k\}$ | 39.33 | 0.2865 | 0.2596 | 88.23 |
| IMDb | {0} | $\{\mathbf{W}_q, \mathbf{W}_v\}$ | 179 | 0.3512 | 0.2702 | 89.56 |
| | {0,11} | $\{\mathbf{W}_q, \mathbf{W}_v\}$ | 5,899 | 0.3597 | 0.2717 | 89.50 |
| | {0, 5, 11} | $\{\mathbf{W}_q, \mathbf{W}_v\}$ | 1,277 | 0.3709 | 0.2727 | 89.49 |
| | {0,5,11} | $\{\mathbf{W}_k\}$ | 9.605 | 0.3642 | 0.2719 | 89.49 |

Table 2: RoBERTa-base model's performance on GLUE tasks, condition number of the NTK, train loss, and evaluation loss at one snapshot of the training at 10-th epoch. LoRA with $r = 8$ is used for fine-tuning. The condition number is calculated as $\kappa(\mathbf{K} + \sigma\mathbf{I}) = \frac{\lambda_{\max}(\mathbf{K}) + \sigma}{\lambda_{\min}(\mathbf{K}) + \sigma}$, and $\sigma = 1e^{-4}$ is fixed among all tasks.

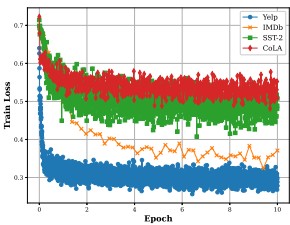 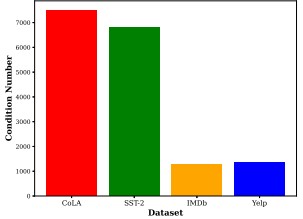 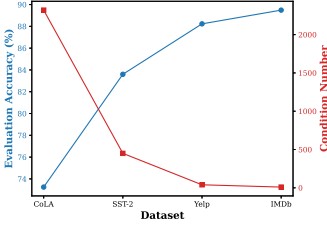

(a) Train loss over 10 epochs

(b) Condition number

(c) Evaluation accuracy and Condition number

Figure 2: (a)-(b) Illustrate the positive correlation between the convergence rate of optimization steps of LoRA over 10 epochs and $\kappa(\mathbf{K} + \sigma\mathbf{I})$ of NTK at initialization. $\{\mathbf{W}_q, \mathbf{W}_v\}$ of layers $\{0, 5, 11\}$ are fine-tuned. (c) Illustrates the negative correlation between evaluation accuracy after 10 epochs of training and the condition number of NTK. LoRA with $r = 8$ is used to fine-tune $\{\mathbf{W}_k\}$ of the layers $\{0, 5, 11\}$.

in Theorems 2 and 4. Notably, the overall accuracy remains largely unaffected by the regularization, and its fluctuations are within the error margin for $\lambda$ up to 10.

The key takeaways from Table 1 are that, as long as we don't overregularize ($\lambda$ not too large, e.g., smaller than 50), we simultaneously: (1) encourage the model not to deviate (the direct objective of regularization), (2) encourage the NTK solution and the regularized fine-tuning solutions to become closer (the indirect main objective of regularization), and (3) do not sacrifice much on performance. We conjecture that the reason we don't sacrifice much on performance is that the fine-tuning solution is already relatively close to the initial model and that there are equivalently good NTK solutions in the same neighborhood. (The choice of $\lambda$ is similar to searching for solutions within sequentially smaller neighborhoods around the initialization, as $\lambda$ increases.) The trifecta of easy implementation, no performance sacrifice, and theoretical compatibility with the NTK framework makes our proposed regularization an especially attractive choice.

## 6.3 NTK Evaluation

We verify our proposition that by calculating the condition number of the NTK matrix for the LoRA model at initialization, we can predict the generalization error, including evaluation loss and accuracy. Table 2 presents training loss, evaluation loss, accuracy, condition number of the NTK before fine-tuning, based on the snapshot at epoch 10. We vary LoRA parameters across different layers ($\{0\}$, $\{11\}$, $\{0,11\}$, $\{0,5,11\}$) for query and value parameters, and layers $\{0,5,11\}$ for key parameters, across various tasks and datasets. We collected $\mathbf{X} = [\mathbf{x}_1, \mathbf{x}_2, \ldots, \mathbf{x}_n]^\top$ using $n = 32$ samples, randomly selected from the training datasets and computed $\mathbf{k}(\mathbf{x}_i, \mathbf{x}_j)$ with respect to trainable parameters, $\mathbf{A}$ and $\mathbf{B}$ of LoRA. The final empirical NTK matrix is $\mathbf{K}(\mathbf{X}, \mathbf{X}) \in \mathbb{R}^{32 \times 32}$. Our work is not specifically studying optimal sketching of the kernel matrix; however, in Appendix I, we empirically illustrate that our numerical results are robust to the choice of NTK samples. Note that the number of samples used for calculation of the empirical NTK is orders of magnitude smaller than the training dataset for sampling. This shows that the results remain valid even with a sketch of the full kernel. This finding highlights the great potential of kernel methods for large language models (LLMs), particularly in terms of efficiency.

At first glance, in Table 2, the highest condition number does not correspond to the lowest accuracy. However, one may notice that this is an issue primarily when the nature (query/value vs. key) and number of parameters change . When this is the case, the effective complexity of the fine-tuned model changes, and since the NTK parameter $\sigma$ is inversely proportional to the complexity, it affects the regularized condition number. In the table, we have fixed $\sigma$ and so this change is not being reflected and it is only fair to compare instead cases where the nature and number of parameters are the same, e.g., fixing a layer and varying tasks among CoLA, SST2, IMDB, YeLP. The correlation is perfect for $\{0, 11\}$, but also mostly tracks for other fixed choices. This also holds when varying the same number of layers like $\{0, 7\}, \{0, 11\}$ for CoLA and SST2 in Fig. 3. It would be very interesting to factor the change of $\sigma$ into the layer selection methodology, but that is a great direction for future research. More extensive numerical results for decoder-only models, such as GPT-2 and OPT-125M, are provided in Appendix J.

Training time and NTK calculation time are reported in Table 3 for fine-tuning $\{\mathbf{W}_k\}$ in layers $\{0, 5, 11\}$. For this scenario, the number of total trainable parameters (TTPs) is 0.628M, and the number of chosen NTK parameters is 36.8k. As shown in the table, fine-tuning, even with just 10 epochs, has significantly higher computational overhead than computing the NTK. This finding supports the advantage of the present approach in terms of time complexity when comparing the risks of different datasets without training. Additionally, since Yelp and IMDb are larger datasets, it is evident that fine-tuning on them requires more time compared to the others. Figure 2(a)-(b) illustrates the positive correlation between the condition number of the NTK matrix at initialization and training loss for different tasks. In all datasets, the attention parameters of layers $\{0, 5, 11\}$ are fine-tuned, and evaluation accuracy was reported. Although for CoLA, it is customary to report Matthew's correlation coefficients [27], we adhere to reporting the evaluation accuracy for all tasks in Figure 2(c), to maintain consistency in the evaluation metric across different datasets. Figure 2(c) starkly illustrates an inverse relationship between the condition number of the NTK and the model's evaluation accuracy. In our experiments, we observed that $\lambda_{\min}(\mathbf{K})$ is almost always close to zero and the regularized condition number, $\kappa(\mathbf{K} + \sigma\mathbf{I})$, is tracing the spectral norm or $\lambda_{\max}(\mathbf{K})$. For instance, the CoLA task, which exhibits the highest training loss, also shows the largest condition number. This suggests that by computing the NTK matrix before training, we can identify which tasks are well-conditioned, i.e., a lower condition number indicates lower training and evaluation loss.

Figure 3 in Appendix L evaluates the bounds of Theorem 7, where $\boldsymbol{\theta}$ represents the weights $\{\mathbf{W}_k\}$ of layer $\{0\}$, and $\hat{\boldsymbol{\theta}}$ denotes the candidate layers. The empirical risk ratio and maximum eigenvalue ratio closely follow each other in most cases. Theorem 4 conveys that the spectrum of the NTK directly affects the empirical risk. Therefore, we propose to select the layers based on the eigenvalues they induce in the NTK. For instance, as seen in Figure 3, for the CoLA dataset, the eigenvalue analysis at-initialization

| Dataset | Fine-tuning Time | NTK Calculation Time |
|---------|------------------|----------------------|
| CoLA    | 187              | 33                   |
| SST-2   | 794              | 63                   |
| Yelp    | 46,096           | 245                  |
| IMDb    | 1,541            | 55                   |

Table 3: Fine-tuning time(s), NTK calculation time(s), $\{\mathbf{W}_k\}$ of layers $\{0, 5, 11\}$ are fine-tuned. In all datasets, only 32 random samples from the training set are used calculating the NTK.

suggests that layers $\{0, 7\}$ should be selected for fine-tuning, and it also shows the least empirical risk compared to others. Similarly, for SST-2, $\{0, 5, 11\}$ is the optimal choice by eigenvalues and is once again consistent with the empirical risk after fine-tuning.

## 7 Conclusion and Limitations

In this paper, we tackled the challenge of guiding design decisions in parameter-efficient fine-tuning by offering insights through linearization. Fine-tuned models are often implicitly close to the pre-trained model. We made this proximity explicit and showed that it results in a model that remains close to the linearization of the pre-trained model. We used the linear model's NTK regression formulation to show that the NTK kernel spectrum can be a predictor of fine-tuning performance, at-initialization (before actual tuning). We then analyzed how this spectrum is affected through layer selection and showed how this can experimentally guide the decision of which layers to tune, saving valuable time and computational resources. Some of the limitations of the theory are that it applies to squared loss and gradient descent, though the experiments show that the insights carry to other losses (such as cross-entropy) and stochastic solvers (such as AdamW). The experiments on RoBERTa with LoRA across multiple datasets (GLUE, IMDb, and Yelp tasks) demonstrate reasonable correspondence between theory and practice, despite some theoretical assumptions differing from experimental conditions. Discussion on the implications of the assumptions is provided in Appendix M.

## Acknowledgments and Disclosure of Funding

This paper is based upon work supported in part by the National Science Foundation, through the NSF CAREER Program under Award No. CCF-2146334 (From Rare Events to Competitive Learning Algorithms), the NSF HDR TRIPODS Phase II Program under Award No. ECCS-2217023 (IDEAL Institute). Computational infrastructure was supported in part by the NSF MRI Program Award No. CNS-1828265 (COMPaaS DLV).

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

# A Definitions and Lemmas

In all the following proofs, we apply the notations and definitions below. Given the data matrices $(\mathbf{X}, \mathbf{Y})$, $\mathbf{X} \in \mathbb{R}^{n \times d}$ and $\mathbf{Y} \in \mathbb{R}^{1 \times n}$ we define the models inference at time step $t$ as

$$\mathbf{Y}(t) = f_{\boldsymbol{\theta}_t}(\mathbf{X}) \in \mathbb{R}^{n \times 1}, \tag{23}$$

The gradient $\nabla_{\theta_t} f_{\boldsymbol{\theta}_t}(\mathbf{X})$, by convention, is in $\mathbb{R}^{n \times \dim(\boldsymbol{\theta})}$. The gradient of (3), the mean squared error, is

$$\nabla_{\boldsymbol{\theta}_t} \widetilde{\mathcal{R}}(\boldsymbol{\theta}_t) = 2 \left(\mathbf{Y}(t) - \mathbf{Y}\right)^\top \nabla f_{\boldsymbol{\theta}_t}(\mathbf{X}). \tag{24}$$

On the other hand, the inference of the linearized model is defined as

$$\bar{\mathbf{Y}}(t) = \bar{f}_{\bar{\boldsymbol{\theta}}_t}(\mathbf{X}) = f_{\boldsymbol{\theta}_0}(\mathbf{X}) + \left\langle \nabla f_{\boldsymbol{\theta}_0}(\mathbf{X}), \bar{\boldsymbol{\theta}}_t - \boldsymbol{\theta}_0 \right\rangle, \tag{25}$$

where $\bar{\boldsymbol{\theta}}$ denotes the parameters of the linearized model, and we assume $\bar{\boldsymbol{\theta}}_0 = \boldsymbol{\theta}_0$ and therefore $\bar{f}_{\bar{\boldsymbol{\theta}}_0}(\mathbf{X}) = f_{\boldsymbol{\theta}_0}(\mathbf{X})$.

For all vectors, we use the $\ell_2$-norm. For all matrices, we use the $\ell_2$ induced norm, or spectral norm, defined as $\|A\| = \sup\{\|Av\|_2 : \|v\|_2 \leq 1\}$, which is also equal to $\sup\{\|uA\|_2 : \|u\|_2 \leq 1\}$ and the largest singular value of $A$.

**Lemma 1.** *If for all $\boldsymbol{\theta}$ in a given range, $f_{\boldsymbol{\theta}}(\mathbf{X})$ is $\mathrm{Lip}(f)$-Lipschitz in $\boldsymbol{\theta}$ (w.r.t. the $\ell_2$-norm) and $\nabla f_{\boldsymbol{\theta}}(\mathbf{X})$ is $\mathrm{Lip}(\nabla f)$-Lipschitz in $\boldsymbol{\theta}$ (w.r.t. the $\ell_2$ induced norm), then $\mathbf{k}_{\boldsymbol{\theta}}(\mathbf{X}, \mathbf{X}) = \nabla f_{\boldsymbol{\theta}}(\mathbf{X}) \nabla f_{\boldsymbol{\theta}}(\mathbf{X})^\top \in \mathbb{R}^{n \times n}$ is $\mathrm{Lip}(k)$-Lipschitz in $\boldsymbol{\theta}$ (w.r.t. the $\ell_2$ induced norm), with*

$$\mathrm{Lip}(\mathbf{k}) \leq 2 \, \mathrm{Lip}(f) \, \mathrm{Lip}(\nabla f). \tag{26}$$

*Proof.* First, since $f_{\boldsymbol{\theta}}$ is $\mathrm{Lip}(f)$-Lipschitz, it follows that the $\ell_2$ induced norm of $\nabla f_{\boldsymbol{\theta}}$ is bounded by $\mathrm{Lip}(f)$. To see this, for any $\dim(\boldsymbol{\theta})$-dimensional vector $\mathbf{w}$, introduce the single parametrized family $\boldsymbol{\theta}(\delta) = \boldsymbol{\theta} + \delta \mathbf{w}$. Then $\frac{d}{d\delta} f_{\boldsymbol{\theta}(\delta)} = \nabla f_{\boldsymbol{\theta}(\delta)} \mathbf{w}$. By Lipschitzness, $\|f_{\boldsymbol{\theta}(\delta)} - f_{\boldsymbol{\theta}(0)}\| \leq \mathrm{Lip}(f) \delta \|\mathbf{w}\|$. Taking the limit $\delta \to 0$ we obtain that $\|\nabla f_{\boldsymbol{\theta}} \mathbf{w}\| \leq \mathrm{Lip}(f) \|\mathbf{w}\|$.

Recall that $\mathbf{k}_{\boldsymbol{\theta}}(\mathbf{X}, \mathbf{X})$ is the NTK matrix. Since all arguments of $\mathbf{k}_{\boldsymbol{\theta}}$, $f_{\boldsymbol{\theta}}$, and $\nabla f_{\boldsymbol{\theta}}$ are $\mathbf{X}$, we omit them for clarity. To show that $\mathbf{k}_{\boldsymbol{\theta}}$ is Lipschitz and to compute its Lipschitz constant, given any $n$-dimensional unit vector $\mathbf{v}$, we need to bound

$$\|(\mathbf{k}_{\boldsymbol{\theta}} - \mathbf{k}_{\boldsymbol{\zeta}})\mathbf{v}\| = \| \left(\nabla f_{\boldsymbol{\theta}} \nabla f_{\boldsymbol{\theta}}^\top - \nabla f_{\boldsymbol{\zeta}} \nabla f_{\boldsymbol{\zeta}}^\top\right) \mathbf{v}\|. \tag{27}$$

We write

$$(\mathbf{k}_{\boldsymbol{\theta}} - \mathbf{k}_{\boldsymbol{\zeta}})\mathbf{v} = \nabla f_{\boldsymbol{\theta}} \left(\nabla f_{\boldsymbol{\theta}}^\top - \nabla f_{\boldsymbol{\zeta}}^\top\right) \mathbf{v} + \left(\nabla f_{\boldsymbol{\theta}} - \nabla f_{\boldsymbol{\zeta}}\right) \nabla f_{\boldsymbol{\zeta}}^\top \mathbf{v}, \tag{28}$$

which yields, by the triangle inequality, norm bounds, and Lipschitzness,

$$\|(\mathbf{k}_{\boldsymbol{\theta}} - \mathbf{k}_{\boldsymbol{\zeta}})\mathbf{v}\| \leq \left\|\nabla f_{\boldsymbol{\theta}} \left(\nabla f_{\boldsymbol{\theta}}^\top - \nabla f_{\boldsymbol{\zeta}}^\top\right) \mathbf{v}\right\| + \left\|\left(\nabla f_{\boldsymbol{\theta}} - \nabla f_{\boldsymbol{\zeta}}\right) \nabla f_{\boldsymbol{\zeta}}^\top \mathbf{v}\right\| \tag{29}$$

$$\leq \|\nabla f_{\boldsymbol{\theta}}\| \left\|\left(\nabla f_{\boldsymbol{\theta}}^\top - \nabla f_{\boldsymbol{\zeta}}^\top\right) \mathbf{v}\right\| + \|\nabla f_{\boldsymbol{\theta}} - \nabla f_{\boldsymbol{\zeta}}\| \|\nabla f_{\boldsymbol{\zeta}}^\top \mathbf{v}\| \tag{30}$$

$$\leq \mathrm{Lip}(f) \|\nabla f_{\boldsymbol{\theta}} - \nabla f_{\boldsymbol{\zeta}}\| + \mathrm{Lip}(\nabla f) \|\boldsymbol{\theta} - \boldsymbol{\zeta}\| \|\nabla f_{\boldsymbol{\zeta}}\| \tag{31}$$

$$\leq \mathrm{Lip}(f) \mathrm{Lip}(\nabla f) \|\boldsymbol{\theta} - \boldsymbol{\zeta}\| + \mathrm{Lip}(\nabla f) \|\boldsymbol{\theta} - \boldsymbol{\zeta}\| \mathrm{Lip}(f) \tag{32}$$

$$= 2 \mathrm{Lip}(f) \mathrm{Lip}(\nabla f) \|\boldsymbol{\theta} - \boldsymbol{\zeta}\|, \tag{33}$$

which proves the claim. $\qquad\square$

**Lemma 2.** *The NTK matrix $\mathbf{K} \in \mathbb{R}^{n \times n}$, defined by $[\mathbf{K}]_{i,j} = \mathbf{k}(\mathbf{x}_i, \mathbf{y}_j)$, is positive semidefinite.*

*Proof.* For all $\mathbf{v} \in \mathbb{R}^n$, we have $\mathbf{v}^\top \mathbf{K} \mathbf{v} = \mathbf{v}^\top \nabla f_{\boldsymbol{\theta}} \nabla f_{\boldsymbol{\theta}}^\top \mathbf{v} = \|\nabla f_{\boldsymbol{\theta}}^\top \mathbf{v}\| \geq 0$. Therefore, $\mathbf{K}$ is positive semi-definite. $\qquad\square$

**Lemma 3.** *Let $\boldsymbol{\theta}_t$ be the gradient flow limit of the regularized fine-tuning gradient descent described in (8). If we assume that $\lambda$ switches at most countably often and denote its instantaneous value by $\lambda_t$, then $\boldsymbol{\theta}_t$ satisfies the following differential equation:*

$$\frac{d}{dt} \boldsymbol{\theta}_t = -\nabla \widetilde{\mathcal{R}}(\boldsymbol{\theta}_t)^\top - \lambda_t \left(\boldsymbol{\theta}_t - \boldsymbol{\theta}_0\right). \tag{34}$$

*Proof.* Subtracting $\boldsymbol{\theta}_0$ from both sides of (5) yields

$$\boldsymbol{\theta}_t - \boldsymbol{\theta}_0 = \widetilde{\boldsymbol{\theta}}_t - \boldsymbol{\theta}_0 - \lambda\eta\left(\widetilde{\boldsymbol{\theta}}_t - \boldsymbol{\theta}_0\right), \tag{35}$$

or equivalently, (8) becomes

$$\boldsymbol{\theta}_t = \boldsymbol{\theta}_0 + (1 - \lambda_t\,\eta)\left(\widetilde{\boldsymbol{\theta}}_t - \boldsymbol{\theta}_0\right), \tag{36}$$

where

$$\lambda_t = \begin{cases} \lambda & \text{if } \nabla_{\boldsymbol{\theta}_t}\widetilde{\mathcal{R}}(\boldsymbol{\theta}_t)\,(\boldsymbol{\theta}_t - \boldsymbol{\theta}_0) \geq 0, \\ 0 & \text{otherwise.} \end{cases} \tag{37}$$

On the other hand, by combining (4) and (5) for $t + 1$ we have

$$\boldsymbol{\theta}_{t+1} = \boldsymbol{\theta}_t - \eta\nabla\widetilde{\mathcal{R}}(\boldsymbol{\theta}_t)^\top - \lambda_{t+1}\,\eta\left(\widetilde{\boldsymbol{\theta}}_{t+1} - \boldsymbol{\theta}_0\right), \tag{38}$$

$$= \boldsymbol{\theta}_t - \eta\nabla\widetilde{\mathcal{R}}(\boldsymbol{\theta}_t)^\top - \frac{\lambda_{t+1}\,\eta}{1 - \lambda_{t+1}\,\eta}\,(\boldsymbol{\theta}_{t+1} - \boldsymbol{\theta}_0), \tag{39}$$

where we replaced $\widetilde{\boldsymbol{\theta}}_{t+1} - \boldsymbol{\theta}_0 = \frac{1}{1 - \lambda_{t+1}\,\eta}\,(\boldsymbol{\theta}_{t+1} - \boldsymbol{\theta}_0)$ from (36) in (38) to obtain (39). By rearranging the terms we have

$$\boldsymbol{\theta}_{t+1} = (1 - \lambda_{t+1}\,\eta)\left(\boldsymbol{\theta}_t - \eta\nabla\widetilde{\mathcal{R}}(\boldsymbol{\theta}_t)^\top + \frac{\lambda_{t+1}\,\eta}{1 - \lambda_{t+1}\,\eta}\boldsymbol{\theta}_0\right), \tag{40}$$

$$= (1 - \lambda_{t+1}\,\eta)\boldsymbol{\theta}_t - \eta(1 - \lambda_{t+1}\,\eta)\nabla\widetilde{\mathcal{R}}(\boldsymbol{\theta}_t)^\top + \lambda_{t+1}\,\eta\,\boldsymbol{\theta}_0. \tag{41}$$

Subtracting $\boldsymbol{\theta}_t$ from both sides and division by $\eta$ yields

$$\frac{\boldsymbol{\theta}_{t+1} - \boldsymbol{\theta}_t}{\eta} = -\lambda_{t+1}\boldsymbol{\theta}_t - (1 - \lambda_{t+1}\,\eta)\nabla\widetilde{\mathcal{R}}(\boldsymbol{\theta}_t)^\top + \lambda_{t+1}\boldsymbol{\theta}_0, \tag{42}$$

$$\frac{d}{dt}\,\boldsymbol{\theta}_t = -\nabla\widetilde{\mathcal{R}}(\boldsymbol{\theta}_t)^\top - \lambda_t\,(\boldsymbol{\theta}_t - \boldsymbol{\theta}_0). \tag{43}$$

To obtain the last equation, we take the limit as $\eta \to 0$ from both sides such that $\frac{\boldsymbol{\theta}_{t+1} - \boldsymbol{\theta}_t}{\eta} \to \frac{d\boldsymbol{\theta}(t)}{dt}$ and $1 - \lambda\eta \to 1$. $\qquad\square$

# B Proof of Theorem 1

From Lemma 3, we reproduce (34), assuming for the moment a constant $\lambda$:

$$\frac{d}{dt}\,\boldsymbol{\theta}_t = -\nabla\widetilde{\mathcal{R}}(\boldsymbol{\theta}_t)^\top - \lambda\,(\boldsymbol{\theta}_t - \boldsymbol{\theta}_0). \tag{44}$$

Using chain rule on (23), and substituting for $\nabla\widetilde{\mathcal{R}}(\boldsymbol{\theta}_t)$ using (24), we have

$$\frac{d}{dt}\,\mathbf{Y}(t) = \nabla f_{\boldsymbol{\theta}_t}(\mathbf{X})\frac{d}{dt}\,\boldsymbol{\theta}_t,$$
$$= -2\,\nabla f_{\boldsymbol{\theta}_t}(\mathbf{X})\,\nabla f_{\boldsymbol{\theta}_t}^\top(\mathbf{X})\,(\mathbf{Y}(t) - \mathbf{Y}) - \lambda\,\nabla f_{\boldsymbol{\theta}_t}(\mathbf{X})\,(\boldsymbol{\theta}_t - \boldsymbol{\theta}_0),$$
$$= -\left(2\,\mathbf{k}_t\,(\mathbf{Y}(t) - \mathbf{Y}) + \lambda\,\nabla f_{\boldsymbol{\theta}_t}(\mathbf{X})\,(\boldsymbol{\theta}_t - \boldsymbol{\theta}_0)\right), \tag{45}$$

To reduce clutter, we again remove the explicit dependency of the kernel on input and denote $\mathbf{k}_t(\mathbf{X}, \mathbf{X}) = \nabla f_{\boldsymbol{\theta}_t}(\mathbf{X})\,\nabla f_{\boldsymbol{\theta}_t}^\top(\mathbf{X})$ by $\mathbf{k}_t$, hereafter. By replacing $\mathbf{Y}(t) = f_{\boldsymbol{\theta}_t}(\mathbf{X})$ and (45) in below equation, we have

$$\frac{1}{2}\frac{d}{dt}\|\mathbf{Y}(t) - \mathbf{Y}\|_2^2 = \frac{1}{2}\frac{d}{dt}\left((\mathbf{Y}(t) - \mathbf{Y})^\top(\mathbf{Y}(t) - \mathbf{Y})\right) = (\mathbf{Y}(t) - \mathbf{Y})^\top\frac{d}{dt}\mathbf{Y}(t), \tag{46}$$

$$= -2(\mathbf{Y}(t) - \mathbf{Y})^\top\mathbf{k}_t\,(\mathbf{Y}(t) - \mathbf{Y}) - \lambda\,(\mathbf{Y}(t) - \mathbf{Y})^\top\nabla f_{\boldsymbol{\theta}_t}(\mathbf{X})\,(\boldsymbol{\theta}_t - \boldsymbol{\theta}_0), \tag{47}$$

$$= -2\langle\mathbf{k}_t\,(\mathbf{Y}(t) - \mathbf{Y}), (\mathbf{Y}(t) - \mathbf{Y})\rangle - \lambda\,(\mathbf{Y}(t) - \mathbf{Y})^\top\nabla f_{\boldsymbol{\theta}_t}(\mathbf{X})\,(\boldsymbol{\theta}_t - \boldsymbol{\theta}_0). \tag{48}$$

Finally,

$$\frac{1}{2}\frac{d}{dt}\|\mathbf{Y}(t) - \mathbf{Y}\|_2^2 \;=\; -2\langle \mathbf{k}_t\,(\mathbf{Y}(t) - \mathbf{Y}),\,(\mathbf{Y}(t) - \mathbf{Y})\rangle - \frac{\lambda}{2}\,\nabla_{\boldsymbol{\theta}}\widetilde{\mathcal{R}}(\boldsymbol{\theta}_t)\,(\boldsymbol{\theta}_t - \boldsymbol{\theta}_0). \qquad (49)$$

Since $\mathbf{k}_t = \nabla f_{\boldsymbol{\theta}_t}(\mathbf{X})\nabla f_{\boldsymbol{\theta}_t}^\top(\mathbf{X})$ is positive semidefinite, we have $\langle \mathbf{k}_t\,(\mathbf{Y}(t) - \mathbf{Y}),\,(\mathbf{Y}(t) - \mathbf{Y})\rangle \geq 0$. To ensure that the error $\|\mathbf{Y}(t) - \mathbf{Y}\|$ does not increase over time, it is sufficient that the gradient $\nabla_{\boldsymbol{\theta}}\widetilde{\mathcal{R}}(\boldsymbol{\theta}_t)$ is aligned with the parameter update direction, i.e.,

$$\nabla_{\boldsymbol{\theta}}\widetilde{\mathcal{R}}(\boldsymbol{\theta}_t)(\boldsymbol{\theta}_t - \boldsymbol{\theta}_0) \geq 0.$$

If this condition is not satisfied, then we should set $\lambda = 0$. This condition is aligned with the selective $\ell_2$ regularization introduced in [9].

## C  Proof of Theorem 2

Once again, use (3), to substitute $\nabla_{\boldsymbol{\theta}}\widetilde{\mathcal{R}}(\boldsymbol{\theta}_t) = 2\,(\mathbf{Y}(t) - \mathbf{Y})^\top\,\nabla f_{\boldsymbol{\theta}_t}(\mathbf{X})$ in (34), yielding

$$\frac{d}{dt}\,\boldsymbol{\theta}_t = -\,2\,\nabla f_{\boldsymbol{\theta}_t}(\mathbf{X})^\top\,(\mathbf{Y}(t) - \mathbf{Y}) \;-\; \lambda_t\,(\boldsymbol{\theta}_t - \boldsymbol{\theta}_0). \qquad (50)$$

Let $\mathbf{u}_t = \boldsymbol{\theta}_t - \boldsymbol{\theta}_0$ and $w(t) = \|\mathbf{u}_t\|_2$. The dynamics can be rewritten as

$$\frac{d}{dt}\,\mathbf{u}_t = -\,2\,\nabla f_{\boldsymbol{\theta}_t}(\mathbf{X})^\top\,(\mathbf{Y}(t) - \mathbf{Y}) \;-\; \lambda_t\,\mathbf{u}_t. \qquad (51)$$

Since $\mathbf{u}_t$ is continuous, $w(t)$ is a.e. differentiable and, for $\mathbf{u}_t \neq 0$,

$$\dot{w}(t) = \frac{\mathbf{u}_t^\top}{\|\mathbf{u}_t\|_2}\frac{d}{dt}\mathbf{u}_t. \qquad (52)$$

Substituting (51) into (52) and writing the unit vector $\hat{\mathbf{u}}_t = \mathbf{u}_t/\|\mathbf{u}_t\|_2$,

$$\dot{w}(t) = -\,2\,\hat{\mathbf{u}}_t^\top\nabla f_{\boldsymbol{\theta}_t}(\mathbf{X})^\top\,(\mathbf{Y}(t) - \mathbf{Y}) \;-\; \lambda_t\,w(t). \qquad (53)$$

By Cauchy–Schwarz and recalling that the $\ell_2$ induced norm $\|\nabla f_{\boldsymbol{\theta}_t}(\mathbf{X})\| \leq \mathrm{Lip}(f)$ (see the proof of Lemma 1),

$$-\hat{\mathbf{u}}_t^\top\nabla f_{\boldsymbol{\theta}_t}(\mathbf{X})^\top\,(\mathbf{Y}(t) - \mathbf{Y}) \;\leq\; \|\nabla f_{\boldsymbol{\theta}_t}(\mathbf{X})\|\,\|\mathbf{Y}(t) - \mathbf{Y}\|_2 \;\leq\; \mathrm{Lip}(f)\,\|\mathbf{Y}(t) - \mathbf{Y}\|_2. \qquad (54)$$

Therefore, we have

$$\dot{w}(t) \;\leq\; -\lambda_t\,w(t) \;+\; 2\,\mathrm{Lip}(f)\,\|\mathbf{Y}(t) - \mathbf{Y}\|_2, \qquad (55)$$

$$\dot{w}(t) \;+\; \lambda_t\,w(t) \;\leq\; 2\,\mathrm{Lip}(f)\,\|\mathbf{Y}(t) - \mathbf{Y}\|_2. \qquad (56)$$

Let

$$\Lambda_t \;=\; \int_0^t \lambda_\tau\,d\tau. \qquad (57)$$

We multiply (56) by the integrating factor $e^{\Lambda_t}$ and use the product rule

$$\frac{d}{dt}\big(e^{\Lambda_t}w(t)\big) \;\leq\; 2\,\mathrm{Lip}(f)\,e^{\Lambda_t}\,\|\mathbf{Y}(t) - \mathbf{Y}\|_2. \qquad (58)$$

We integrate (58) from 0 to $t$ as

$$e^{\Lambda_t}w(t) - w(0) \;\leq\; 2\,\mathrm{Lip}(f)\int_0^t e^{\Lambda_s}\,\|\mathbf{Y}(s) - \mathbf{Y}\|_2\,ds. \qquad (59)$$

Then we divide by $e^{\Lambda_t}$ to obtain

$$w(t) \;\leq\; e^{-\Lambda_t}\,w(0) \;+\; 2\,\mathrm{Lip}(f)\int_0^t e^{-(\Lambda_t - \Lambda_s)}\,\|\mathbf{Y}(s) - \mathbf{Y}\|_2\,ds, \qquad (60)$$

$$\leq\; e^{-\Lambda_t}\,w(0) \;+\; 2\,\mathrm{Lip}(f)\|\mathbf{Y}(0) - \mathbf{Y}\|_2\int_0^t e^{-(\Lambda_t - \Lambda_s)}\,ds, \qquad (61)$$

where the last inequality follows from Theorem 1.

**Special case 1.** If $\lambda_t = \lambda > 0$, then $\Lambda_t = \lambda t$ therefore we have

$$\|\mathbf{u}_t\|_2 \ \leq \ e^{-\lambda t}\|\mathbf{u}_0\|_2 \ + \ 2\operatorname{Lip}(f)\|\mathbf{Y}(0) - \mathbf{Y}\|_2 \int_0^t e^{-\lambda(t-s)}\, ds, \tag{62}$$

$$\leq \ e^{-\lambda t}\|\mathbf{u}_0\|_2 \ + \ 2\operatorname{Lip}(f)\|\mathbf{Y}(0) - \mathbf{Y}\|_2 \frac{1 - e^{-\lambda t}}{\lambda}. \tag{63}$$

Since $\mathbf{u}_t = \boldsymbol{\theta}_t - \boldsymbol{\theta}_0$, $\|\mathbf{u}_0\|_2 = 0$, and $\mathbf{Y}(0) = f_{\boldsymbol{\theta}_0}(\mathbf{X})$, we have

$$\|\boldsymbol{\theta}_t - \boldsymbol{\theta}_0\|_2 \ \leq \ 2\operatorname{Lip}(f)\,\|f_{\boldsymbol{\theta}_0}(\mathbf{X}) - \mathbf{Y}\|_2 \frac{1 - e^{-\lambda t}}{\lambda}, \tag{64}$$

$$\limsup_{t\to\infty} \|\boldsymbol{\theta}_t - \boldsymbol{\theta}_0\|_2 \ \leq \ \frac{2\operatorname{Lip}(f)\,\|f_{\boldsymbol{\theta}_0}(\mathbf{X}) - \mathbf{Y}\|_2}{\lambda}. \tag{65}$$

**Special case 2.** If $\lambda_t = 0$, from (61), we have

$$\|\mathbf{u}_t\|_2 \leq \|\mathbf{u}_0\|_2 + 2\operatorname{Lip}(f)\,\|\mathbf{Y}(0) - \mathbf{Y}\|_2 \int_0^t e^0\, ds. \tag{66}$$

Therefore,

$$\|\boldsymbol{\theta}_t - \boldsymbol{\theta}_0\|_2 \leq 2\operatorname{Lip}(f)\,\|f_{\boldsymbol{\theta}_0}(\mathbf{X}) - \mathbf{Y}\|_2\, t. \tag{67}$$

## D   Proof of Theorem 3

For simplicity, we consider the lambda constant case to prove Theorems 3 and 4. For $\mathbf{Y}(t) = f_{\boldsymbol{\theta}_t}(\mathbf{X})$ and $\bar{\mathbf{Y}}(t) = \bar{f}_{\bar{\boldsymbol{\theta}}_t}(\mathbf{X})$, we define $\Delta(t) = \|\mathbf{Y}(t) - \bar{\mathbf{Y}}(t)\|_2$. We have

$$\frac{1}{2}\frac{d}{dt}\Delta(t)^2 = \frac{1}{2}\frac{d}{dt}\|\mathbf{Y}(t) - \bar{\mathbf{Y}}(t)\|_2^2 \tag{68}$$

$$= \frac{1}{2}\langle \mathbf{Y}'(t) - \bar{\mathbf{Y}}'(t), \mathbf{Y}(t) - \bar{\mathbf{Y}}(t)\rangle + \frac{1}{2}\langle \mathbf{Y}(t) - \bar{\mathbf{Y}}(t), \mathbf{Y}'(t) - \bar{\mathbf{Y}}'(t)\rangle$$

$$= \langle \mathbf{Y}'(t) - \bar{\mathbf{Y}}'(t), \mathbf{Y}(t) - \bar{\mathbf{Y}}(t)\rangle$$

$$= \big\langle -\mathbf{k}_t 2\,(\mathbf{Y}(t) - \mathbf{Y}) - \lambda\,\nabla f_{\boldsymbol{\theta}_t}(\mathbf{X})\,(\boldsymbol{\theta}_t - \boldsymbol{\theta}_0) + \mathbf{k}_0\, 2\,(\bar{\mathbf{Y}}(t) - \mathbf{Y}), \mathbf{Y}(t) - \bar{\mathbf{Y}}(t)\big\rangle,$$

where the last equality follows from (45) in the proof of Theorem 1. We simplify the following term in the last equation as

$$-\mathbf{k}_t 2\,(\mathbf{Y}(t) - \mathbf{Y}) + \mathbf{k}_0\, 2\,(\bar{\mathbf{Y}}(t) - \mathbf{Y}) \tag{69}$$

$$= -\mathbf{k}_t 2\,(\mathbf{Y}(t) - \mathbf{Y}) + \mathbf{k}_0 2\,(\mathbf{Y}(t) - \mathbf{Y}) - \mathbf{k}_0 2\,(\mathbf{Y}(t) - \mathbf{Y}) + \mathbf{k}_0\, 2\,(\bar{\mathbf{Y}}(t) - \mathbf{Y})$$

$$= (\mathbf{k}_0 - \mathbf{k}_t)\, 2\,(\mathbf{Y}(t) - \mathbf{Y}) + \mathbf{k}_0\, \big(2\,(\bar{\mathbf{Y}}(t) - \mathbf{Y}) - 2\,(\mathbf{Y}(t) - \mathbf{Y})\big),$$

where in the first equality we added and subtracted $\mathbf{k}_0 2\,(\mathbf{Y}(t) - \mathbf{Y})$. Substituting (69) in (68) yields

$$\frac{1}{2}\frac{d}{dt}\Delta(t)^2 = \langle (\mathbf{k}_0 - \mathbf{k}_t)\, 2\,(\mathbf{Y}(t) - \mathbf{Y}) - \lambda\,\nabla f_{\boldsymbol{\theta}_t}(\mathbf{X})\,(\boldsymbol{\theta}_t - \boldsymbol{\theta}_0), \mathbf{Y}(t) - \bar{\mathbf{Y}}(t)\rangle$$

$$+ \big\langle \mathbf{k}_0\, \big(2\,(\bar{\mathbf{Y}}(t) - \mathbf{Y}) - 2\,(\mathbf{Y}(t) - \mathbf{Y})\big), \mathbf{Y}(t) - \bar{\mathbf{Y}}(t)\big\rangle, \tag{70}$$

since $\mathbf{k}_0 = \nabla f_{\boldsymbol{\theta}_0}(\mathbf{X})\,\nabla f_{\boldsymbol{\theta}_0}(\mathbf{X})^\top$ is positive semidefinite, the second term on the left hand side is non-positive, i.e.,

$$\big\langle \mathbf{k}_0\, \big(2\,(\bar{\mathbf{Y}}(t) - \mathbf{Y}) - 2\,(\mathbf{Y}(t) - \mathbf{Y})\big), \mathbf{Y}(t) - \bar{\mathbf{Y}}(t)\big\rangle$$

$$= -2(\mathbf{Y}(t) - \bar{\mathbf{Y}}(t))^\top \mathbf{k}_0(\mathbf{Y}(t) - \bar{\mathbf{Y}}(t)) \leq 0. \tag{71}$$

As a result of (99) and (71), we have

$$\frac{1}{2}\frac{d}{dt}\Delta(t)^2 \leq \langle (\mathbf{k}_0 - \mathbf{k}_t)\, 2\,(\mathbf{Y}(t) - \mathbf{Y}) - \lambda\,\nabla f_{\boldsymbol{\theta}_t}(\mathbf{X})\,(\boldsymbol{\theta}_t - \boldsymbol{\theta}_0), \mathbf{Y}(t) - \bar{\mathbf{Y}}(t)\rangle. \tag{72}$$

Noting that $\frac{1}{2}\frac{d}{dt}\Delta(t)^2 = \Delta(t)\frac{d\Delta(t)}{dt}$ and taking norms on both sides of the above, then clearly using (36), we have

$$\frac{d}{dt}\Delta(t) = \| (\mathbf{k}_0 - \mathbf{k}_t) \, 2 \, (\mathbf{Y}(t) - \mathbf{Y}) - \lambda \, \nabla f_{\boldsymbol{\theta}_t}(\mathbf{X}) \, (\boldsymbol{\theta}_t - \boldsymbol{\theta}_0) \| \tag{73}$$

$$\leq \| (\mathbf{k}_0 - \mathbf{k}_t) \, 2 \, (\mathbf{Y}(t) - \mathbf{Y}) \| + \lambda \| \nabla f_{\boldsymbol{\theta}_t}(\mathbf{X}) \, (\boldsymbol{\theta}_t - \boldsymbol{\theta}_0) \| \tag{74}$$

$$\leq \mathrm{Lip}(\mathbf{k}) \| (\boldsymbol{\theta}_t - \boldsymbol{\theta}_0) \| \| 2 \, (\mathbf{Y}(t) - \mathbf{Y}) \| + \lambda \, \mathrm{Lip}(f) \, \| \boldsymbol{\theta}_t - \boldsymbol{\theta}_0 \| \tag{75}$$

$$\leq 2\mathrm{Lip}(\mathbf{k}) \| (\boldsymbol{\theta}_t - \boldsymbol{\theta}_0) \| \| \mathbf{Y}(0) - \mathbf{Y} \| + \lambda \, \mathrm{Lip}(f) \, \| \boldsymbol{\theta}_t - \boldsymbol{\theta}_0 \| \tag{76}$$

$$\leq 4 \, \mathrm{Lip}(\mathbf{k})\mathrm{Lip}(f) \| \mathbf{Y}(0) - \mathbf{Y} \|^2 \frac{\left(1 - e^{-\lambda t}\right)}{\lambda} \tag{77}$$
$$+ 2 \, \mathrm{Lip}^2(f) \| \mathbf{Y}(0) - \mathbf{Y} \| \left(1 - e^{-\lambda t}\right)$$

$$\leq -2 \, \mathrm{Lip}(f) \| \mathbf{Y}(0) - \mathbf{Y} \| \left( \frac{2}{\lambda} \mathrm{Lip}(\mathbf{k}) \| \mathbf{Y}(0) - \mathbf{Y} \| + \mathrm{Lip}(f) \right) e^{-\lambda t} \tag{78}$$
$$+ 2 \, \mathrm{Lip}(f) \| \mathbf{Y}(0) - \mathbf{Y} \| \left( \frac{2}{\lambda} \mathrm{Lip}(\mathbf{k}) \| \mathbf{Y}(0) - \mathbf{Y} \| + \mathrm{Lip}(f) \right).$$

(75) follows from Lipschitz properties of $f_{\boldsymbol{\theta}}(\mathbf{X})$ and $\mathbf{k}_t$ which was shown in Lemma 1. Due to the non-increasing property of $\| \mathbf{Y}(t) - \mathbf{Y} \|$ shown in Theorem 1, we replaced $\mathbf{Y}(t)$ with $\mathbf{Y}(0)$ to obtain the upper bound. We also used (10) in Theorem 2 to obtain the last inequality.

Using Lemma 1 and (78), we have

$$\frac{d}{dt}\Delta(t) \leq b - be^{-\lambda t}, \tag{79}$$

where

$$b = 2 \, \mathrm{Lip}(f)^2 \| \mathbf{Y}(0) - \mathbf{Y} \| \left( \frac{4}{\lambda} \mathrm{Lip}(\nabla f) \| \mathbf{Y}(0) - \mathbf{Y} \| + 1 \right). \tag{80}$$

By taking the integral of both sides of (79), we obtain

$$\Delta(t) \leq b \left( t + \frac{1}{\lambda} e^{-\lambda t} - \frac{1}{\lambda} \right). \tag{81}$$

Consequently, we have (12).

# E  Proof of Theorem 4

From Theorem 2, we have

$$\| \boldsymbol{\theta}_t - \boldsymbol{\theta}_0 \| \leq \frac{2 \, \| \mathbf{Y}(0) - \mathbf{Y} \| \, \mathrm{Lip}(f)}{\lambda} \left( 1 - e^{-\lambda t} \right) \tag{82}$$

In order for the regularizer to satisfy the Lipschitz continuity assumptions, mainly that $\| \boldsymbol{\theta} - \boldsymbol{\theta}_0 \| \leq r$, this shows that there are two phases of behavior depending on how large $\lambda$ is. The threshold is given by:

$$\lambda_\circ = \frac{2 \| \mathbf{Y}(0) - \mathbf{Y} \| \, \mathrm{Lip}(f)}{r}$$

In particular, if $\lambda \geq \lambda_\circ$, then $\boldsymbol{\theta}_t$ remains in the $r$-ball around $\boldsymbol{\theta}_0$ for all $t$. Otherwise, it remains in this ball only as long as

$$t \leq \frac{1}{\lambda} \ln \frac{1}{1 - \lambda/\lambda_\circ}.$$

Based on Theorem 3, we get that

$$\Delta(t) \leq 2 \, \mathrm{Lip}(f)^2 \| \mathbf{Y}(0) - \mathbf{Y} \| \left( \frac{4}{\lambda} \mathrm{Lip}(\nabla f) \| \mathbf{Y}(0) - \mathbf{Y} \| + 1 \right) \left( t + \frac{1}{\lambda} e^{-\lambda t} - \frac{1}{\lambda} \right).$$

When $\lambda \geq \lambda_\circ$, $\Delta(t)$ grows roughly linearly, with coefficient given by:

$$\Delta(t) \leq 2 \, \mathrm{Lip}(f) \| \mathbf{Y}(0) - \mathbf{Y} \| \, (2r\mathrm{Lip}(\nabla f) + \mathrm{Lip}(f)) \, t.$$

# F  Proof of Theorem 5

Let $\mathbf{U\Sigma U}^\top$ denote the eigenvalue decomposition of $\mathbf{K}\left(\mathbf{X}, \mathbf{X}\right)$, where $\mathbf{\Sigma} = \mathrm{Diag}\left(\lambda_{\min}(\mathbf{K}), \ldots, \lambda_{\max}(\mathbf{K})\right)$ and $\mathbf{U}^\top \mathbf{U} = \mathbf{I}$. Due to the proximity of the regularized fine-tuned model to the pretrained model, which promotes linearity, we have

$$
\begin{aligned}
\mathcal{R}(\boldsymbol{\theta}) &= \frac{1}{n}\sum_{i=1}^{n}\mathcal{L}(f_{\boldsymbol{\theta}}(\mathbf{x}_i), \mathbf{y}_i) \approx \frac{1}{n}\sum_{i=1}^{n}\mathcal{L}(\bar{f}_{\bar{\boldsymbol{\theta}}}(\mathbf{x}_i), \mathbf{y}_i)\\
&= \frac{1}{n}\sum_{i=1}^{n}\left\| y_i - \mathbf{K}\left(\mathbf{x}_i, \mathbf{X}\right)\left[\mathbf{K}\left(\mathbf{X}, \mathbf{X}\right) + \sigma\mathbf{I}\right]^{-1}\mathbf{Y}\right\|_2^2\\
&= \frac{1}{n}\left\| \mathbf{Y} - \mathbf{K}\left(\mathbf{X}, \mathbf{X}\right)\left(\mathbf{K}\left(\mathbf{X}, \mathbf{X}\right) + \sigma\mathbf{I}\right)^{-1}\mathbf{Y}\right\|_2^2\\
&= \frac{1}{n}\left\| \left(\mathbf{I} - \mathbf{K}\left(\mathbf{X}, \mathbf{X}\right)\left(\mathbf{K}\left(\mathbf{X}, \mathbf{X}\right) + \sigma\mathbf{I}\right)^{-1}\right)\mathbf{Y}\right\|_2^2\\
&= \frac{1}{n}\left\| \left(\mathbf{I} - \mathbf{U\Sigma U}^\top(\mathbf{U\Sigma U}^\top + \sigma\mathbf{I})^{-1}\right)\mathbf{Y}\right\|_2^2\\
&= \frac{1}{n}\left\| \left(\mathbf{I} - \mathbf{U\Sigma}(\mathbf{\Sigma} + \sigma\mathbf{I})^{-1}\mathbf{U}^\top\right)\mathbf{Y}\right\|_2^2\\
&= \frac{1}{n}\left\| \mathbf{U}\left(\mathbf{I} - \mathbf{\Sigma}(\mathbf{\Sigma} + \sigma\mathbf{I})^{-1}\right)\mathbf{U}^\top\mathbf{Y}\right\|_2^2\\
&= \frac{1}{n}\left\| \left(\mathbf{I} - \mathbf{\Sigma}(\mathbf{\Sigma} + \sigma\mathbf{I})^{-1}\right)\mathbf{U}^\top\mathbf{Y}\right\|_2^2.
\end{aligned}
\tag{83}
$$

Since $\mathbf{I} - \mathbf{\Sigma}(\mathbf{\Sigma} + \sigma\mathbf{I})^{-1}$ is a diagonal matrix, we have

$$
\lambda_{\min}\left(\mathbf{I} - \mathbf{\Sigma}(\mathbf{\Sigma} + \sigma\mathbf{I})^{-1}\right)^2\|\mathbf{U}^\top\mathbf{Y}\|_2^2 < \mathcal{R}(\boldsymbol{\theta}) < \lambda_{\max}(\mathbf{I} - \mathbf{\Sigma}(\mathbf{\Sigma} + \sigma\mathbf{I})^{-1})^2\|\mathbf{U}^\top\mathbf{Y}\|_2^2.
\tag{84}
$$

Noting that

$$
\lambda_{\min}\left(\mathbf{I} - \mathbf{\Sigma}(\mathbf{\Sigma} + \sigma\mathbf{I})^{-1}\right) = \frac{\sigma}{\sigma + \lambda_{\max}\left(\mathbf{K}\right)},
\tag{85}
$$

$$
\lambda_{\max}\left(\mathbf{I} - \mathbf{\Sigma}(\mathbf{\Sigma} + \sigma\mathbf{I})^{-1}\right) = \frac{\sigma}{\sigma + \lambda_{\min}\left(\mathbf{K}\right)},
\tag{86}
$$

we have

$$
\frac{\sigma^2\|\mathbf{Y}\|_2^2}{(\sigma + \lambda_{\max}(\mathbf{K}))^2} \leq \mathcal{R}(\boldsymbol{\theta}) \leq \frac{\sigma^2\|\mathbf{Y}\|_2^2}{(\sigma + \lambda_{\min}(\mathbf{K}))^2}
\tag{87}
$$

which is also shown in [28].

# G  Proof of Theorem 6

Since $\mathbf{K}$ is positive semidefinite (see Lemma 2), we have

$$
\mathbf{K} + \mathbf{S} = \mathbf{K}^{1/2}\left(\mathbf{I} + \mathbf{K}^{-1/2}\mathbf{S}\,\mathbf{K}^{-1/2}\right)\mathbf{K}^{1/2}.
\tag{88}
$$

Considering $\eta = \|\mathbf{K}^{-1/2}\mathbf{S}\,\mathbf{K}^{-1/2}\|$, we have

$$
-\eta\mathbf{I} \leq \mathbf{K}^{-1/2}\mathbf{S}\,\mathbf{K}^{-1/2} \leq \eta\mathbf{I}.
\tag{89}
$$

Consequently,

$$
(1 - \eta)\mathbf{I} \leq \mathbf{I} + \mathbf{K}^{-1/2}\mathbf{S}\,\mathbf{K}^{-1/2} \leq (1 + \eta)\mathbf{I},
\tag{90}
$$

and, from (88) and (90), we obtain

$$
(1 - \eta)\mathbf{K} \leq \mathbf{K} + \mathbf{S} \leq (1 + \eta)\mathbf{K}.
\tag{91}
$$

Note that for any two positive semidefinite matrices $\mathbf{A}$ and $\mathbf{B}$, if $\mathbf{A} \leq \mathbf{B}$, then $\lambda_i(\mathbf{A}) \leq \lambda_i(\mathbf{B})$. Therefore, (91) is equivalent to (20) [29].

## H   Proof of Theorem 7

From Theorem 5, we already know

$$\frac{(\lambda_{\min}(\mathbf{K}) + \sigma)^2}{\sigma^2 \|\mathbf{Y}\|^2} \leq \frac{1}{\mathcal{R}(\boldsymbol{\theta})} \leq \frac{(\lambda_{\max}(\mathbf{K}) + \sigma)^2}{\sigma^2 \|\mathbf{Y}\|^2}. \tag{92}$$

Theorem 6, states that

$$\lambda_{\max}(\mathbf{K}) + \sigma = \lambda_{\max}(\mathbf{K} + \sigma\mathbf{I}) \leq \frac{\lambda_{\max}(\mathbf{K} + \mathbf{S} + \sigma\mathbf{I})}{1 - \eta} \tag{93}$$

and

$$\lambda_{\min}(\mathbf{K}) + \sigma = \frac{\lambda_{\max}(\mathbf{K} + \sigma\mathbf{I})}{\kappa(\mathbf{K} + \sigma\mathbf{I})} \geq \frac{\lambda_{\max}(\mathbf{K} + \mathbf{S} + \sigma\mathbf{I})}{\kappa(\mathbf{K} + \sigma\mathbf{I})(1 + \eta)}. \tag{94}$$

Therefore, we have

$$\frac{\lambda_{\max}(\mathbf{K} + \mathbf{S} + \sigma\mathbf{I})}{\sigma\|\mathbf{Y}\|\kappa(\mathbf{K} + \sigma\mathbf{I})(1 + \eta)} \leq \frac{1}{\mathcal{R}(\boldsymbol{\theta})^{\frac{1}{2}}} \leq \frac{\lambda_{\max}(\mathbf{K} + \mathbf{S} + \sigma\mathbf{I})}{\sigma\|\mathbf{Y}\|(1 - \eta)}. \tag{95}$$

On the other hand, it follows from Theorem 5 that

$$\frac{\sigma\|\mathbf{Y}\|}{\lambda_{\max}(\mathbf{K} + \mathbf{S}) + \sigma} \leq \mathcal{R}(\boldsymbol{\theta} \cup \hat{\boldsymbol{\theta}})^{\frac{1}{2}} \leq \frac{\sigma\|\mathbf{Y}\|}{\lambda_{\min}(\mathbf{K} + \mathbf{S}) + \sigma}, \tag{96}$$

and, from Theorem 6 that

$$\lambda_{\min}(\mathbf{K} + \mathbf{S}) + \sigma \geq (1 - \eta)\lambda_{\min}(\mathbf{K} + \sigma\mathbf{I})$$
$$= \frac{(1 - \eta)\lambda_{\max}(\mathbf{K} + \sigma\mathbf{I})}{\kappa(\mathbf{K} + \sigma\mathbf{I})},$$
$$\lambda_{\max}(\mathbf{K} + \mathbf{S}) + \sigma \leq (1 + \eta)\lambda_{\max}(\mathbf{K} + \sigma\mathbf{I}). \tag{97}$$

From (96) and (97), we conclude

$$\frac{\sigma\|\mathbf{Y}\|}{(1 + \eta)\lambda_{\max}(\mathbf{K} + \sigma\mathbf{I})} \leq \mathcal{R}(\boldsymbol{\theta} \cup \hat{\boldsymbol{\theta}})^{\frac{1}{2}} \leq \frac{\sigma\|\mathbf{Y}\|\kappa(\mathbf{K} + \sigma\mathbf{I})}{(1 - \eta)\lambda_{\max}(\mathbf{K} + \sigma\mathbf{I})}. \tag{98}$$

Considering that $\kappa(\mathbf{K} + \sigma\mathbf{I}) \leq c$, the inequalities (98) and (95) imply

$$\frac{\lambda_{\max}(\mathbf{K} + \mathbf{S} + \sigma\mathbf{I})}{c(1 + \eta)^2 \lambda_{\max}(\mathbf{K} + \sigma\mathbf{I})} \leq \left(\frac{\mathcal{R}(\boldsymbol{\theta} \cup \hat{\boldsymbol{\theta}})}{\mathcal{R}(\boldsymbol{\theta})}\right)^{\frac{1}{2}} \leq \frac{c\lambda_{\max}(\mathbf{K} + \mathbf{S} + \sigma\mathbf{I})}{(1 - \eta)^2 \lambda_{\max}(\mathbf{K} + \sigma\mathbf{I})}. \tag{99}$$

Note that $(1 - \eta)^2 \leq (1 + \eta)^{-2}$. By defining $a = \frac{c}{(1-\eta)^2}$, we can rewrite (99) in the desired form:

$$\frac{\lambda_{\max}(\mathbf{K} + \mathbf{S} + \sigma\mathbf{I})}{a\lambda_{\max}(\mathbf{K} + \sigma\mathbf{I})} \leq \left(\frac{\mathcal{R}(\boldsymbol{\theta} \cup \hat{\boldsymbol{\theta}})}{\mathcal{R}(\boldsymbol{\theta})}\right)^{\frac{1}{2}} \leq \frac{a\lambda_{\max}(\mathbf{K} + \mathbf{S} + \sigma\mathbf{I})}{\lambda_{\max}(\mathbf{K} + \sigma\mathbf{I})}. \tag{100}$$

# I  Robustness to Sampling for Estimating the NTK

We empirically illustrate that the eigenvalues of the NTK are robust to the choice of NTK samples. Sketching of the kernel matrices is studied in the literature, for instance, in [30]. In [17], which also modeled fine-tuning as an NTK regression problem, the number of NTK samples is fixed to 16 and 64 (see Table 2 of [17]).

| Random Seed | $\lambda_{\min}$ | $\lambda_{\max}$ | Condition Number |
|:---:|:---:|:---:|:---:|
| 42 | $2.04 \times 10^{-6}$ | 0.0030 | 32.43 |
| 123 | $6.45 \times 10^{-7}$ | 0.0035 | 36.85 |
| 7 | $7.37 \times 10^{-8}$ | 0.0025 | 26.22 |
| 99 | $3.19 \times 10^{-9}$ | 0.0024 | 25.82 |
| 2024 | $6.78 \times 10^{-8}$ | 0.0025 | 26.86 |

Table 4: Condition numbers and eigenvalues for different random seeds. $\mathbf{W}_k$ of layer 11 is fine-tuned.

# J  Additional Experiments

We fine-tuned decoder-only models GPT-2 and OPT-125M for 10 epochs using the Adam optimizer. LoRA with $r = 8$ is used to fine-tune $\mathbf{W}_k$ of the layers $\{0, 5, 11\}$. The negative correlation between evaluation accuracy after 10 epochs of training and the condition number of the NTK is illustrated below. In GPT-2, Yelp with the lowest condition number, possesses the highest accuracy, and in OPT-125M, IMDb and Yelp, with higher accuracies, have lower condition numbers than the other tasks.

| Model | Dataset | Eval Accuracy (%) | Condition Number |
|:---:|:---|:---:|:---:|
| GPT-2 | CoLA | 71.0 | 83 |
| | IMDb | 87.4 | 36 |
| | Yelp | 87.6 | 35 |
| OPT-125m | SST-2 | 64.8 | 910 |
| | CoLA | 69.1 | 720 |
| | IMDb | 70.0 | 210 |
| | Yelp | 82.1 | 310 |

Table 5: Evaluation results for GPT-2 and OPT-125M across different datasets.

# K  Lipschitzness

To empirically estimate Lipschitzness for all models in balls around the initial model $\boldsymbol{\theta}_0$, we need to choose models $\boldsymbol{\theta}$ in this ball and calculate the Lipschitz constants for each using pairs of feature points $(\mathbf{x}, \mathbf{x}')$. Since it's not feasible to choose all models and all pairs, we have to sample them reasonably. In the table below, we report on $\boldsymbol{\theta}$ sampled uniformly on consecutive spherical shells with increasing radius and $(\mathbf{x}, \mathbf{x}')$ samples from pairs of training data points. We calculate the estimated Lipschitz constant for each $\boldsymbol{\theta}$ and report both the average and maximum estimates, for each radius. The accurate Lipschitz constant $\text{Lip}(f)$, technically, is the maximum. However, the average is also meaningful in that it represents more typical practical behavior.

We conducted the following procedure. We first capture the maximum fluctuation, $R_{\max}$, in parameters $\boldsymbol{\theta}$ as in Table 6 .

| Dataset | Layer | $R_{\max}$ |
|:---|:---|:---:|
| CoLA | $\mathbf{W}_k \in \{0, 5, 11\}$ | 0.174 |
| CoLA | $\mathbf{W}_k \in \{0, 11\}$ | 0.189 |

Table 6: $R_{\max} = \text{Max}|\boldsymbol{\theta} - \boldsymbol{\theta}_0|$ during 10 epochs of fine-tuning.

**Algorithm 1** Computation of $L_{\text{avg}}$ and $L_{\text{upper}}$ vs. $r$

---

**Input:** Set $\mathcal{S}$ of data samples $(\mathbf{x}, \mathbf{x}')$
1: Initialize list $L_{\text{max}}$ (indexed by $\boldsymbol{\theta}$)
2: Initialize lists $L_{\text{avg}}, L_{\text{upper}}$ (indexed by $r$)
3: **for** $r = 0$ to $R_{\text{max}}$ **do**
4:     Generate a new set $T(r)$ of $n_T$ models using

$$\boldsymbol{\theta} = \boldsymbol{\theta}_0 + \text{distortion}(r), \quad \text{where} \quad \text{distortion}(r) = \frac{rv}{\|v\|}, \quad v \sim \mathcal{N}(0, 1)$$

5:     **for all** $\boldsymbol{\theta} \in T(r)$ **do**
6:         Initialize empty list $L_{\text{list}}$
7:         Set model params to $\boldsymbol{\theta}$
8:         **for all** $(\mathbf{x}, \mathbf{x}') \in \mathcal{S}$ **do**
9:             Compute $\dfrac{\|f_{\boldsymbol{\theta}}(\mathbf{x}') - f_{\boldsymbol{\theta}}(\mathbf{x})\|}{\|\mathbf{x}' - \mathbf{x}\|}$ and append to $L_{\text{list}}$
10:         **end for**
11:         Append $\max(L_{\text{list}})$ to $L_{\text{max}}$
12:     **end for**
13:     Append $\text{mean}(L_{\text{max}})$ to $L_{\text{avg}}$
14:     Append $\max(L_{\text{max}})$ to $L_{\text{upper}}$
15: **end for**
**Output:** $L_{\text{avg}}$ and $L_{\text{upper}}$ vs. $r$

---

Then we used $R_{\text{max}}$ to calculate the Lipschitz continuity as in Algorithm 1. We used $|\mathcal{S}| = 1000$ pairs of data points, $n_T = 100$ models, i.e., 100 different $\boldsymbol{\theta}$s, with $R_{\text{max}} = 0.174$ according to Table 6, 10 steps to vary distortion $r$ in step 3 of the algorithm, and regularization value $\lambda = 5$. Both the average, $L_{\text{avg}}$, and upper bound, $L_{\text{upper}}$, Lipschitz constants show a gradual and consistent increase as $r$ grows, indicating that the model's sensitivity to perturbations increases mildly with distance from the original parameters $\theta_0$. On average, the model remains stable under small to moderate distortions.

| $r$ | $L_{\text{avg}}$ | $L_{\text{upper}}$ |
|---|---|---|
| 0.000000 | 0.020724 | 0.020724 |
| 0.019333 | 0.020725 | 0.020764 |
| 0.038667 | 0.020726 | 0.020803 |
| 0.058000 | 0.020727 | 0.020843 |
| 0.077333 | 0.020727 | 0.020883 |
| 0.096667 | 0.020728 | 0.020923 |
| 0.116000 | 0.020729 | 0.020962 |
| 0.135333 | 0.020730 | 0.021002 |
| 0.154667 | 0.020731 | 0.021042 |
| 0.174000 | 0.020732 | 0.021081 |

Table 7: Lipschitz ratio of the model with selected layers $\{0, 5, 11\}$ and parameter type key ($\mathbf{W}_k$) for 1000 pairs of data samples $(\mathbf{x}, \mathbf{x}')$ with $R_{\text{max}} = 0.174$.

| $r$ | $L_{\text{avg}}$ | $L_{\text{upper}}$ |
|---|---|---|
| 0.000000 | 0.014479 | 0.014479 |
| 0.021000 | 0.014480 | 0.014505 |
| 0.042000 | 0.014482 | 0.014530 |
| 0.063000 | 0.014483 | 0.014556 |
| 0.084000 | 0.014484 | 0.014582 |
| 0.105000 | 0.014486 | 0.014608 |
| 0.126000 | 0.014487 | 0.014634 |
| 0.147000 | 0.014488 | 0.014660 |
| 0.168000 | 0.014490 | 0.014686 |
| 0.189000 | 0.014491 | 0.014712 |

Table 8: Lipschitz ratio of the model with selected layers $\{0, 11\}$ and parameter type key ($\mathbf{W}_k$) for 100 pairs of data samples $(\mathbf{x}, \mathbf{x}')$ with $R_{\text{max}} = 0.189$.

# L  Layer Selection Algorithm

Building upon the foundation established by Theorem 7, Corollary 1, and Figure 3, we distill our proposed methodologies into the following PEFT layer selection strategy informed by our spectral perturbation bounds.

---

**Algorithm 2** Trainable Parameter Selection via Spectral Perturbation

---

    **Input:** Pretrained parameters $\boldsymbol{\theta}$; scalar $\sigma > 0$; training samples $\mathbf{X} = [\mathbf{x}_1, \ldots, \mathbf{x}_n]^\top$; candidates parameter subsets $\{\hat{\boldsymbol{\theta}}^{(1)}, \ldots, \hat{\boldsymbol{\theta}}^{(L)}\}$; $\mathcal{C} = \{1, \ldots, L\}$
1: Compute base NTK matrix $\mathbf{K} \in \mathbb{R}^{n \times n}$.
2: **for** $l = 1, \ldots, L$ **do**
3:     Compute kernel contribution $\mathbf{S}_l$ for candidate $\hat{\boldsymbol{\theta}}^{(l)}$.
4: **end for**
5: **for** each subset $C \in \mathcal{C}$ **do**
6:     Compute combined kernel $\mathbf{S}^C \leftarrow \sum_{l \in C} \mathbf{S}_l$.
7:     Compute spectral ratio
$$r_c \leftarrow \frac{\lambda_{\max}\big(\mathbf{K} + \mathbf{S}^C + \sigma \mathbf{I}\big)}{\lambda_{\max}(\mathbf{K} + \sigma \mathbf{I})}.$$
8: **end for**
9: Select $C^* \leftarrow \underset{C}{\arg\min}\, r_c$.
    **Output:** Selected parameters $\{\hat{\boldsymbol{\theta}}^{(l)} : l \in C^*\}$.

---

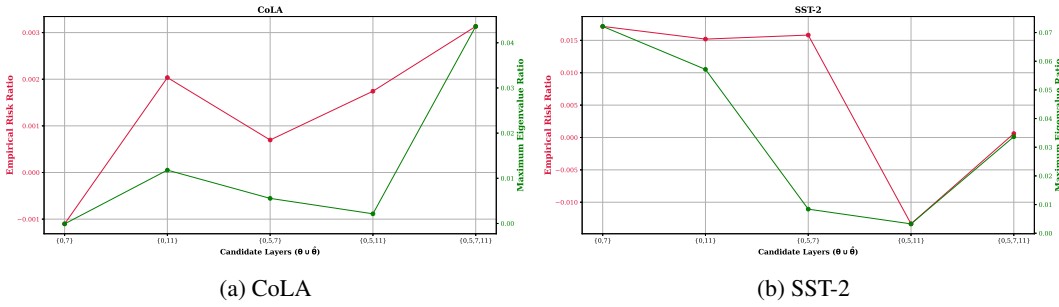

(a) CoLA            (b) SST-2

Figure 3: Empirical risk ratio $\log\left(\frac{\mathcal{R}(\boldsymbol{\theta} \cup \hat{\boldsymbol{\theta}})}{\mathcal{R}(\boldsymbol{\theta})}\right)$ and maximum eigenvalue ratio $\log\left(\frac{\lambda_{\max}(\mathbf{K} + \mathbf{S} + \sigma \mathbf{I})}{\lambda_{\max}(\mathbf{K} + \sigma \mathbf{I})}\right)$ are used to evaluate the impact of candidate layers. Here, $\boldsymbol{\theta}$ is fixed as the weights $\{\mathbf{W}_k\}$ of layer $\{0\}$, while $\hat{\boldsymbol{\theta}}$ represents the candidate layers. The horizontal axis represents the combination of layer $\{0\}$ and different candidate layers.

## M  Discussion and Implication of Assumptions

- **Stochastic vs. deterministic gradient descent:** Our theoretical derivations are based on deterministic gradient descent, while the AdamW optimizer, which is stochastic, is used in our experiments. Our theory is designed to make a few key aspects precise: (1) that it is possible to directly elicit NTK regime behavior through regularization and (2) that, once we operate in the NTK regime, the spectrum of the kernel determines training performance. Note that (2) does not rely on any particular optimization scheme. As for (1), while it is true that the noisy gradient aspect of stochastic approaches is not part of the theory, gradient descent/flow captures the local-search approach of gradient-based methods, including SGD and its variants. Applying selective regularization (Eq. (8)) is as straightforward as applying gradient clipping with SGD and Adam. The bounds of Theorems 3 and 4 transfer to stochastic variants through standard bounds that link their respective trajectories. (These typically need strong convexity assumptions on the loss functions and bounds on the variance of the stochastic gradient [31].

- **Regularized vs. non-regularized fine-tuning** It is crucial to emphasize that our linearity analysis only applies to the explicitly regularized variant of fine-tuning that we present. Even though we believe that this regularization may sometimes exist implicitly during fine-tuning, establishing bounds between non-regularized fine-tuning and linearized fine-tuning remains outside the scope of this paper and is an interesting avenue for future investigation.

- **Cross-entropy vs. mean squared loss:** Our theoretical results are derived for the squared loss, while our experiments succeed with cross-entropy. According to [31], we provide intuitions on why our theoretical insights generalize well to different loss functions. Note that optimizing cross-entropy is equivalent to optimizing KL-divergence, and KL-divergence and squared loss are intimately related, given that they're both Bregman divergences. For two outputs that are close to each other, i.e., in the high-accuracy regime, the KL-divergence behaves very similarly to the squared loss [32]. As such, while the theory doesn't apply directly to cross-entropy, the behavior that we demonstrate qualitatively supports the behavior that we experimentally observe with cross-entropy.

- **Lipschitzness** The Lipschitz continuity assumption in Theorems 2- 4 is not limiting. Almost every architecture and training methodology used in practice limits the complexity of the networks. Of particular relevance to our paper are the robust variants of low-rank adaptation, which explicitly enforce this kind of Lipschitz condition [33]. In Appendix K, we empirically estimate the Lipschitz constant for models in balls around the initial model.

## N  Hyperparameters

| Dataset | CoLA | SST-2 | Yelp | IMDb |
|---|---|---|---|---|
| Optimizer | | AdamW | | |
| Warmup Ratio | | 0.06 | | |
| LR Schedule | | Linear | | |
| Max Sequence Length | | 512 | | |
| LoRA Rank $r$ | | 8 | | |
| LoRA $\alpha$ | | 8 | | |
| Number of Epochs | | 10 | | |
| Batch Size | 32 | 16 | 32 | 16 |
| Learning Rate | 4e-4 | 5e-4 | 4e-4 | 4e-4 |

Table 9: Hyperparameters used for the RoBERTa-base model on various benchmarks, including GLUE (CoLA, SST-2), Yelp, and IMDb.

