# OpenReview forum: "Linearization Explains Fine-Tuning in Large Language Models"
_NeurIPS.cc/2025/Conference — NeurIPS 2025 poster_

### Official Review · Reviewer_xfCG · 2025-06-18

**Clarity:** 2
**Significance:** 2
**Originality:** 2
**Rating:** 4
**Confidence:** 3

**Summary:**

The paper studies fine-tuning through linearization and NTK regression. The paper shows under an explicit regularized fine-tuning process, the behavior of model dynamics is close to a linearized model, which leads to the approximation by NTK. They then analyze how the eigenvalue spectrum of the NTK is related to the model performance and validated through empirical experiments.

**Questions:**

1. It is not clear what is the implication of Theorem 2. Comparing the bound for regularized fine-tuning (9) and conventional fine-tuning (10), the bound (9) could be larger than (10) given the exponential dependence on $t$ versus linear dependence on $t$. In this case, how such regularization $\lambda$ would manifest lazy training?

2. In Theorem 4, what is $N$? And it seems by setting $t = O(1/\lambda \log(N/\text{Lip}^2))$ and requiring $\lambda \geq O(2N/(r \text{Lip}))$ would bring $t = O(r\text{Lip} \log(N/\text{Lip}^2 )/N)$. Could you comment on whether such requirement on $t$ is restrictive or not?

3. In Theorem 4, can you comment on how large $\lambda$ should be in order for the fine-tuning to be approximated by NTK regression?

4. In Theorem 5, what is $\mathcal{R}$? It seems only $\widetilde{\mathcal{R}}$ is defined in (3). I think $\mathcal{R}$ is defined in terms of (14), is it correct?

5. I do not quite understand how Table 2 correlates to your propositions. Section 6.3 does not have sufficient discussions of the results in Table 2. I do not understand why "lower condition number indicates lower training and evaluation loss". In your Theorem 5, the condition number seems only to suggest the range of your empirical risk.

6. It seems the main benefit of the propositions is to determine what layers need to fine-tuned. However, I did not see much discussion in this regard, namely how can we determine the layer to be fine-tuned in practice. It is better to write explicitly an algorithm for such purpose.

**Ethical Concerns:**

["NO or VERY MINOR ethics concerns only"]

**Final Justification:**

My main concerns on the theoretical results are addressed. More discussions are expected in the revised version.

**Limitations:**

Yes.

**Quality:**

2

**Strengths And Weaknesses:**

The paper provides some insights on how fine-tuning is related to NTK and how the eigenvalue spectrum of NTK predicts the model performance. Such an insight could be useful for determining what layers to be fine-tuned by evaluating the corresponding NTK spectrum. However, analyzing fine-tuning dynamics with NTK is not new and has been studied in e.g., [1]. And it is unclear what is the main procedure for fine-tuning with layer selection via NTK. Also see weaknesses in the questions section.


[1] LoRA training in the NTK regime has no spurious local minima. ICML 2024.

---

> ### Author Rebuttal · Authors · 2025-07-31
>
> Dear Reviewer xfCG,
>
> Thank you for taking the time to review our paper and for the incisive yet constructive remarks. We appreciate it a lot. We would like to address your comments below and hope that you will consider revising your assessment upward.
>
> ---
>
> **Distinction from [1]** The work in [1] explores the loss landscape in the NTK regime; however, it does not determine under what conditions fine-tuning falls within this regime. In contrast, our work proves—both theoretically and empirically—that although fine-tuning is not necessarily in the NTK (linearization) regime, it can be constrained to this regime by regularizing the fine-tuning process towards the pretrained model. Additionally, we provide a spectral perturbation bound for layer selection, which, to the best of our knowledge, is the first of its kind. Therefore, our work is distinct from [1].
>
> ---
>
> **Role of Theorem 2** (Q1) This theorem is primarily a building block for Theorem 3, it simply says that while the solution may deviate from the origin under regularization, we can bound that deviation. As you point out, this may not be the tightest way to bound the deviation, but it is still useful. We could relegate it to a lemma. Theorem 3 and 4 show that we can jump from a deviation from initialization bound (which is the direct  objective of regularization) to a bound on how far the NTK solution and the regularized fine-tuning solutions are  (the indirect but **main** benefit of regularization). This is what allows us to place the approximation of Section 4 on a stronger footing than prior works that assume that such proximity simply holds.
>
> ---
>
> **Theorem 4** (Q2) The analysis Theorem 3 and 4 is based on gradient flow, which is a continuous process  (see Theorem 2.3 in [2]). $t$ represents the continuous time in the gradient flow model. What $N$ represents is a discretization relationship between the continuous time gradient-flow analysis and gradient descent, and is analogous to the number of iterations. We will define $N$ more explicitly in the statement of the Theorem. (To be more technical, one step of gradient descent approximates the gradient flow over a small time interval. The size of this time interval (the "step-size") is related to the properties of the function being optimized, specifically $1/\\text{Lip}(h)^2$ and regularization parameter  $\\lambda$. Moreover the choice of $t=\\frac{1}{\\lambda} \\log(1+\\frac{N}{\\text{Lip}(f)^2})$ results $|\\theta_t-\\theta_0|<r$ and ensures  Lipschitz continuity  in equation (79).)
>
> (Q3) If we sidestep $N$, we can express $t$ directly as a function of $\\lambda$. The result is an inverse dependence, up to the logarithmic term. The takeaway is that the regularization determines the time scale at which the approximation holds, as well as how tight the approximation is. Regarding the exact choice of $\\lambda$, Theorem 4, at first glance, seems to suggest that the bigger the better to bring NTK and regularized trajectories together. However, larger $\\lambda$ means that we effectively force ourselves to remain in a small ball around $\\theta_0$ and may suffer in terms of fine-tuning performance (see $\\lambda=50$ in Table 1). The "sweet spot" tradeoff can be read from Theorem 4 by noting that the right-hand is simply $t$ times the middle parenthesis factor $\\frac{2}{\\lambda} \\textrm{Lip}(\\nabla f) \\Vert f_{\\theta_0}(\\mathbf{x}^\\star)-\\mathbf{y^\\star}\\Vert+1$. Therefore the parenthesis term is the one that governs the accuracy. Since there is a $+1$, there is no point optimizing $\\lambda$ beyond $\\textrm{Lip}(\\nabla f)$ times the largest error of the base model.
>
> ---
>
> **Empirical risk** (Q4) We apologize for this notational mismatch. It is true that in this context we are using (14), but the definition of empirical risk is the same (does not include the Hilbert norm). We will unify the notation across all sections:
>
>  $​​\\mathcal{R}= \\frac{1}{n} \\sum_{i=1}^{n} \\mathcal{L}(f_{\\mathbf{\\theta}}(\\mathbf{x}_i), \\mathbf{x}_i)$
>
> ---
>
> **Condition Number vs. Performance** (Q5) It is true that Theorem 5, taken as it is, does not say that the condition number dictates performance. It does say that the condition number determines the tightness of the bounds (how close the upper and lower bounds are). However, if we assume that the spectrum is mostly stable (say the arithmetic or geometric midpoint of $\\lambda_{\\textsf{min}}$ and $\\lambda_{\\textsf{max}}$ remains roughly constant), then the changes in condition number would indeed dictate the magnitude of the bounds.
>
> It is also true that at first glance, in Table 2, the highest condition number does not correspond to the lowest accuracy. However, one may notice that this is an issue primarily when the nature and number of parameters change (query/value vs. key). When this is the case, the effective complexity of the fine-tuned model changes, and since the NTK parameter $\\sigma$ is inversely proportional to the complexity, it affects the *regularized* condition number. In the table, we have fixed $\\sigma$ and so this change is not being reflected and it is only fair to compare instead cases where the nature and number of parameters is the same, e.g. fixing a layer and varying tasks among CoLA, SST2, IMDB, YeLP, or (the correlation is perfect for $\\{0,11\\}$, but also mostly tracks for other fixed choices), or varying the same number of layers like  $\\{0,7\\}$, $\\{0,11\\}$ for CoLA and SST2 in Figure 3. It would be very interesting to factor the change of $\\sigma$$ into the layer selection methodology, but that is a great direction of future research.
>
> ---
>
> **Main procedure for fine-tuning with layer selection via NTK** (Q6) While we didn't explicitly present it algorithmically, we can use Theorem 7, Corollary 1 and Figure 3 to distil our suggested methodologies into the following PEFT layer selection strategy, using our spectral perturbation bounds:
>
> > ### Algorithm: Trainable Parameter Selection via Spectral Kernel Perturbation
> >
> >**Input:**
> >- Pretrained model parameters $ \\boldsymbol{\\theta} $
> >- $ \\sigma > 0 $
> >- Training samples $ \mathbf{X} = [x_1, \\dots, x_n]^{\\top} $
> >- Candidate parameter subsets $ \{ \hat{\\boldsymbol{\\theta}}^{(1)}, \\dots, \\hat{\\boldsymbol{\\theta}}^{(L)} \} $  and $\mathcal{C}=\\{1,\\ldots,L\\}$
> >
> >**Step 1:** Compute base NTK matrix $ \\mathbf{K} \\in \\mathbb{R}^{n \\times n} $
> >
> >**Step 2:** For each candidate $ \\hat{\\boldsymbol{\\theta}}^{(l)}$, compute kernel contribution $ \\mathbf{S}_l $
> >
> >**Step 3:** For each subset $ C \\in \\mathcal{C}$:
> >- Compute combined kernel: $ \mathbf{S}^C = \\sum_{l \in C} \\mathbf{S}_l $
> >- Compute spectral ratio:  $r_c = \frac{\\lambda_{\\max}(\\mathbf{K} + \\mathbf{S}^C + \\sigma >\mathbf{I})}{\\lambda_{\\max}(\\mathbf{K} + \\sigma \\mathbf{I})}$
> >- Select:  $C^* = \\underset{C}{\\arg\\min} \\; r_c $
> >
> > **Output:** Selected parameters  $\\{\\!\\{ \\hat{\boldsymbol{\\theta}}^{(l)} : l \\in C^*\\}\\!\\} $
>
> ---
>
> Thank you once again. We hope you will consider raising your score and we look forward to discussing further!
> Best,
> Authors
>
> ---
>
> [1] Jang, U., Lee, J.D. and Ryu, E.K., 2024, July. LoRA training in the NTK regime has no spurious local minima. In Proceedings of the 41st International Conference on Machine Learning (pp. 21306-21328).
>
> [2] Chizat, L., Oyallon, E. and Bach, F., 2019. On lazy training in differentiable programming. Advances in neural information processing systems, 32.

---

> > ### Comment · Reviewer_xfCG · 2025-08-04
> >
> > I thank the authors for their detailed responses. Please consider including more discussions regarding the concerns on Theorem 2 and 4 as they are essential for interpreting your results. After all, my main concerns are addressed and I am happy to increase my score.

---

> > > ### Author Response · Authors · 2025-08-07
> > > **Thank you**
> > >
> > > We acknowledge that these explanations should have been there from the get go and will certainly include them with the additional space. Thank you for engaging with us and for increasing your (hidden from us) score!

---

### Official Review · Reviewer_Ptpw · 2025-06-26

**Clarity:** 3
**Significance:** 3
**Originality:** 3
**Rating:** 5
**Confidence:** 4

**Summary:**

This paper presents a theoretical framework for understanding Parameter-Efficient Fine-Tuning (PEFT) in large language models (LLMs) through linearization and NTK theory. The authors introduce explicit $\(l_2\)$-distance regularization toward pretrained parameters, showing that this induces "lazy training" dynamics equivalent to NTK regression. Key contributions include: (1) Theoretical bounds on the distance between fine-tuned and linearized models, (2) Characterization of fine-tuning performance via the NTK eigenvalue spectrum, and (3) Spectral perturbation bounds for layer-wise adaptation.

**Questions:**

1. **Loss Function Generalization**: Your theory assumes MSE, but experiments use cross-entropy. Can you discuss whether the NTK-performance correlation holds *theoretically* for cross-entropy? If not, what adjustments are needed?
2. **Methodological Extensions**: Could your spectral perturbation bounds (Thm 7) directly inform a *layer selection algorithm* (e.g., prioritizing layers that minimize $\lambda_{\text{max}}(\mathbf{K} + \mathbf{S})$? Have you tested such an approach?  Moreover, some more novel methodological innovations are preferred. (See Weakness 2)
3. **Regularization Discrepancy**: In Table 1, higher $\lambda$ reduces $\|\theta_t - \theta_0\|$ but inconsistently affects accuracy (e.g., CoLA peaks at $\lambda=5$). How do you reconcile this with the claim that accuracy is "unaffected"?
4. **Condition Number Correlation**: Table 2 shows CoLA’s best accuracy (73.44%) occurs at $\kappa = 7,503$ , which is not the smallest one. What explains cases where low $\kappa$ *does not* yield the best performance? In fact, it seems that the performance is more likely to be *unaffected* by condition numbers, a more precise explanation is favored.
5. Typo: Line 167, *infinitesimally*

**Ethical Concerns:**

["NO or VERY MINOR ethics concerns only"]

**Final Justification:**

All of my concerns have been addressed. I believe this is a work of broad interest for researchers concerning PEFT and NTK analysis on LMs, thus warranting an acceptance. The authors should supplement their rebuttal explanations in their final version of the manuscript.

**Limitations:**

See Weaknesses and Questions.

**Paper Formatting Concerns:**

Line 261 is partially masked by the figure below. Please fix.

**Quality:**

3

**Strengths And Weaknesses:**

Strengths：
1. **Theoretical Rigor**: The paper rigorously formalizes PEFT dynamics through NTK linearization, deriving novel bounds for model deviation (Theorems 2–4) and risk minimization (Theorem 5). The spectral perturbation analysis (Theorems 6–7) provides actionable insights for layer selection.
2. **Practical Utility**: Demonstrates that NTK condition number (computable beforehand) correlates with downstream task performance, enabling efficient layer selection. Table 3 highlights significant time savings.
3. **Innovative Regularization**: The selective $\(l_2\)$-regularization scheme, activated based on gradient alignment, is a nuanced methodological contribution that ensures monotonic loss decrease.
4. **Strong Validation**: Experiments across GLUE, IMDb, and Yelp tasks consistently support theoretical claims, particularly the NTK-performance correlation and regularization efficacy.

Weaknesses:
1. **Loss Function Gap**: Theory relies on **MSE loss**, while experiments use cross-entropy. The generalization of NTK spectral analysis to classification losses remains unproven.
2. **Methodological Novelty**: While offering theoretical insights, the work does **not propose a new algorithm**. The selective regularization is underexplored as a standalone method. Moreover, a linear fine-tune strategy in [11](author's citation number) seems to be further justified by this work, which should be used as a analyzed baseline, or even should be referred to design a new tuning method.
3. **Inconsistent Empirical Claims**:
   - The claim that "*accuracy remains largely unaffected by regularization*" (Sec 6.2) contradicts Table 1 (SST-2 accuracy varies by **3.7%** with $\lambda$, variations are even larger on CoLA).
   - Table 2 shows weak correlation between $\kappa(\mathbf{K} + \sigma\mathbf{I})$ and accuracy.
4. **Limited Scope**: Theoretical bounds assume Lipschitz continuity and small parameter deviations, which may not hold for aggressive fine-tuning.

---

> ### Author Rebuttal · Authors · 2025-07-31
>
> Dear Reviewer Ptpw,
>
> Thank you for taking the time to review our paper. We would like to address your comments below and hope that you will consider revising your assessment upward.
>
> ---
>
> **Loss Function** (Q1) Regarding the adherence to the squared loss in both (1) and (2), note that optimizing cross-entropy is equivalent to optimizing KL-divergence, and KL-divergence and squared-loss are intimately related, given that they're both Bregman divergences. (For two outputs that are close to each other, i.e., in the high-accuracy regime, KL-divergence behaves very similarly to squared loss. [1]) As such, while the theory doesn't apply directly to cross-entropy, the behavior that we demonstrate qualitatively supports the behavior that we experimentally observe with cross-entropy.
>
> ---
>
> **Methodological Novelty** (Q2) While we didn't explicitly present it algorithmically, we can use Theorem 7, Corollary 1 and Figure 3 to distil our suggested methodologies into the following PEFT layer selection strategy, using our spectral perturbation bounds:
>
> > ### Algorithm: Trainable Parameter Selection via Spectral Kernel Perturbation
> >
> >**Input:**
> >- Pretrained model parameters $ \\boldsymbol{\\theta} $
> >- $ \\sigma > 0 $
> >- Training samples $ \mathbf{X} = [x_1, \\dots, x_n]^{\\top} $
> >- Candidate parameter subsets $\\{ \hat{\\boldsymbol{\\theta}}^{(1)}, \\dots, \\hat{\\boldsymbol{\\theta}}^{(L)} \\} $ and $\mathcal{C}=\\{1,\\ldots,L\\}$
> >
> >**Step 1:** Compute base NTK matrix $ \\mathbf{K} \\in \\mathbb{R}^{n \\times n} $
> >
> >**Step 2:** For each candidate $ \\hat{\\boldsymbol{\\theta}}^{(l)}$, compute kernel contribution $ \\mathbf{S}_l $
> >
> >**Step 3:** For each subset $ C \\in \\mathcal{C}$:
> >- Compute combined kernel:  $\\mathbf{S}^C = \\sum_{l \in C} \\mathbf{S}_l $
> >- Compute spectral ratio:   $r_c = \frac{\\lambda_{\\max}(\\mathbf{K} + \\mathbf{S}^C + \\sigma >\mathbf{I})}{\\lambda_{\\max}(\\mathbf{K} + \\sigma \\mathbf{I})}$
> >- Select:  $C^* = \\underset{C}{\\arg\\min} \\; r_c $
> >
> >**Output:** Selected parameters  $\\{\\!\\{ \\hat{\boldsymbol{\\theta}}^{(l)} : l \\in C^*\\}\\!\\} $
>
>
> We believe this is the key novelty in our paper. We will also investigate using [11](paper reference) as baseline. Thank you for the suggestion.
>
> ---
>
> **Regularization vs. Accuracy** (Q3) We would like to correct a misunderstanding regarding Table 1. Accuracies are negatively affected *only* when we overregularize ($\\lambda$ not too large, e.g., smaller than 50). Apart from this, our claim that accuracy is unaffected is true (the fluctuations are within the error margin for $\\lambda$ up to 10. The takeaway from this table is that whereas the *direct* objective of regularization is to encourage the model not to deviate, the *indirect* benefit we get is that it encourages the NTK solution and the regularized fine-tuning solutions to become closer. This is what makes the results on Section 4 apply. The trifecta of easy implementation + no performance sacrifice + enabling the NTK theory hold makes our proposed regularization an attractive option.
>
> ---
>
> **Is Lipschitz continuity limiting?** The general answer is no. Almost every architecture and training methodology used in practice limits the complexity of the networks. Of particular relevance to our paper are the robust variants of low-rank adaptation, which explicitly enforce this kind of Lipschitz conditions. [2]
>
> ---
>
> **Condition Number Correlation** (Q4) It is true that at first glance, in Table 2, the highest condition number does not correspond to the lowest accuracy. However, one may notice that this is an issue primarily when the nature and number of parameters change (query/value vs. key). When this is the case, the effective complexity of the fine-tuned model changes, and since the NTK parameter $\\sigma$ is inversely proportional to the complexity, it affects the *regularized* condition number. In the table, we have fixed $\\sigma$ and so this change is not being reflected and it is only fair to compare instead cases where the nature and number of parameters is the same, e.g. fixing a layer and varying tasks among CoLA, SST2, IMDB, YeLP, or (the correlation is perfect for $\\{0,11\\}$, but also mostly tracks for other fixed choices), or varying the same number of layers like  $\\{0,7\\}$, $\\{0,11\\}$ for CoLA and SST2 in Figure 3. It would be very interesting to factor the change of $\\sigma$$ into the layer selection methodology, but that is a great direction of future research.
>
> ---
>
> Thank you once again. We hope you will consider raising your score and we look forward to discussing further!
> Best,
> Authors
>
> ---
>
> [1] Borade, S. and Zheng, L., 2008, March. Euclidean information theory. In 2008 IEEE International Zurich Seminar on Communications (pp. 14-17).
>
> [2] Savostianova, D., Zangrando, E., Ceruti, G. and Tudisco, F., 2023. Robust low-rank training via approximate orthonormal constraints. Advances in Neural Information Processing Systems, 36, pp.66064-66083.

---

> ### Comment · Reviewer_Ptpw · 2025-08-04
> **Ack of Rebuttal and Further Discussions**
>
> Thanks for the rebuttal. Some of my concerns have been resolved, and here are some further questions.
>
> Towards Q1. Though it is still not that rigorous as the optimization trajectory could vary for MSE v.s. CE, the current explanation is okay because the empirical results seem solid. But I suggest supplementing those explanations in the paper explicitly.
>
> Towards Q2. I am curious whether the proposed method could be integrated into [11], e.g. non-linear/linear finetuning with layer/module selection, and how the determined "optimal" subset of parameters would perform compared to the full (non-)linear FT in [11]. Maybe some additional experimental results could strengthen your empirical contributions.
>
> Towards Q3, Q4. I think the explanation is persuasive and should be discussed in the main text.
>
> Towards Lipschitzness. I reckon that there are works generally assuming the Lipschitzness of NNs during fine-tuning, but I think an empirical visualization/estimation of the upper bound of the Lipschitz constant can make the theory more sound.
>
> It would be great to further supplement those contents, and I would be happy to raise my score.

---

> > ### Comment · Reviewer_Ptpw · 2025-08-06
> > **Still waiting for the response**
> >
> > Sorry for clicking on the mandatory ack too early. Actually, I am still waiting for the response to my further questions from the authors. I have now deleted the mandatory ack, and I will restore it after the discussion.

---

> > > ### Author Response · Authors · 2025-08-07
> > >
> > > Thank you for engaging us in discussion!
> > >
> > > ---
> > >
> > > (Q1) Yes, we will include the relationship between MSE and CE explicitly in the paper. We will also illustrate some results for MSE, to emphasize that the theory applies as is too.
> > >
> > > ---
> > >
> > > (Q2) In [11], fine-tuning is explicitly performed in the NTK regime and that is shown to produce weight disentanglement, which is itself beneficial for task arithmetic (merging/ablating fine-tuned tasks). There are two ways in which one could combine that paper with the present one. (1) The most obvious way  is to replace the tangent space fine-tuning with our regularized variant. A proper study would then strive to show that weight disentanglement is preserved (which is the key, according to that paper, to reap the benefits of task arithmetic). That's of course out of our current scope but, given our results showing that we're close to linearity, it is not a stretch. (2) The next question is whether our spectral perturbation bounds and our layer selection methodology would be useful too. Here, the key hurdle to cross is that our results are framed with respect to *individual* tasks. If we take an end-to-end perspective with task arithmetic, then we would want the layer selection to predict performance on desired *combined* tasks. Here's an idea of how it could be done. For additive combinations, if we assume that adding tasks is tantamount to fine-tuning with multiple datasets at the same time, then the kernel and its spectrum could be estimated using the desired combinations of datasets. Our results would then apply. For subtractive combinations, it may be possible to select layers in such a way that some tasks' predicted performances are higher (the added ones), while others' are lower (the subtracted ones). However, since tasks may interact in complex ways, whether this is the best approach is not clear and more research is needed to make it rigorous. We do, however, very much welcome the suggestion of exploring this!
> > >
> > > ---
> > >
> > > (Q3) and (Q4) Thank you for this. We acknowledge that these explanations should have been there from the get go and will certainly include them with the additional space.
> > >
> > >
> > > ---
> > >
> > > (Lipschitzness) To empirically estimate Lipschitzness for all models in balls around the initial model $\\theta_0$, we need to choose models $\\theta$ in this ball and calculate the Lipschitz constants for each using pairs of feature points $(x,x')$. Since it's not feasible to choose *all* models and *all* pairs, we have to sample these reasonably. In the table below, we report on $\\theta$ sampled uniformly on consecutive spherical shells with increasing radius and $(x,x')$ samples from pairs of training data points. We calculate the estimated Lipschitz constant for each $\\theta$ and report both the average and maximum of the estimates, for each radius. The quantity from the paper, technically, is the maximum. However, the average is also meaningful in that it represents more typical practical behavior. We can include such evaluations as part of the paper's appendix.

---

> ### Author Response · Authors · 2025-08-07
>
> We  conducted the following procedure. We  first capture the maximum fluctuation, $R_{\\text{max}}$, in parameters $\theta$ as below.
>
> | **Dataset** | **Layer**                        | **$R_{\\text{max}}$** |
> |-------------|----------------------------------|-------------------------------------------------|
> | CoLA        | $\\mathbf{W}_k \\in \\{0, 5, 11\\}$  | 0.174                                           |
> | CoLA        | $\\mathbf{W}_k \\in \\{0, 11\\}$     | 0.189                                           |
>
> *Table: $R_{max} = \\text{max} |\\theta - \\theta_0|$ during 10 epochs of fine-tuning.*
>
>
> Then we  used $R_{\\text{max}}$ to calculate the Lipschitz continuity as  below:
>
> > ### Algorithm: Calculating Lipschitz constant
> >**Input:** set $\\mathcal{S}$ of data points $(x,x')$, and model with pre-trained params $\\theta_0$
> >
> > Initialize list $L_{\\text{max}}$ (will be indexed by theta)
> >
> > Initialize list/array $L_{\\text{avg}}$, $L_{\text{upper}}$ (will be indexed by $r$)
> >
> > For $r$ from $0$ to $R_{\\text{max}}$
> >- Create a new set $T(r)$ by generating $n_{T}$ parameters of the form $\\theta = \\theta_0 + \\text{distortion}(r)$ where $ \\text{distortion}(r)=r*v/||v||$, $v \\sim \\mathcal{N}(0,1)$
> >- For $\\theta$ in $T(r)$
> >>- Create an empty list $L_{\\text{list}}$
> >>- Set model params to $\\theta$
> >>- For $(x,x')  \\in \\mathcal{S}$ Calculate and append $||f_{\\theta}(x')-f_{\\theta}(x)||/||x'-x||$ to $L_{\\text{list}}$
> >>- Append $max(L_{\\text{list}})$ to $L_{\\text{max}}$
> >- Append $mean(L_{\\text{max}})$ to $L_{\\text{avg}}$
> >- Append $max(L_{\\text{max}})$ to $L_{\\text{upper}}$
> >
> >**Output:**  $L_\\text{avg}$  and  $L_\\text{upper}$  vs.  $r$
>
>
> |    $r$  |   $L_\\text{avg}$   |  $L_\\text{upper}$  |
> |----------|-----------|-----------|
> | 0.000000 | 0.020724  | 0.020724  |
> | 0.019333 | 0.020725  | 0.020764  |
> | 0.038667 | 0.020726  | 0.020803  |
> | 0.058000 | 0.020727  | 0.020843  |
> | 0.077333 | 0.020727  | 0.020883  |
> | 0.096667 | 0.020728  | 0.020923  |
> | 0.116000 | 0.020729  | 0.020962  |
> | 0.135333 | 0.020730  | 0.021002  |
> | 0.154667 | 0.020731  | 0.021042  |
> | 0.174000 | 0.020732  | 0.021081  |
>
> *Table: Lipschitz ratio of the model with selected layers {$0, 5, 11$} and parameter type *key* for $1000$ pairs of data.*
>
> We use $|\\mathcal{S}| = 1000$ (number of pairs of data points), $n_T = 100$ (models, i.e., 100 different $\\theta$), with $R_{\\max} = |\\theta - \\theta_0| = 0.175$,  for 10 distortions of $r$, and regularizer value $\\lambda = 5$.
> Both the average $L_{\\text{avg}}$ and upper bound $L_{\\text{upper}}$​ of Lipschitz constants show a gradual and consistent increase as $r$ grows, indicating that the model’s sensitivity to perturbations increases mildly with distance from the original parameters $\\theta_{0}$​. On average, the model remains stable under small to moderate distortions.
>
> |    $r$  |   $L_\\text{avg}$   |  $L_\\text{upper}$  |
> |----------|------------|------------|
> | 0.000000 | 0.014479   | 0.014479   |
> | 0.021000 | 0.014480   | 0.014505   |
> | 0.042000 | 0.014482   | 0.014530   |
> | 0.063000 | 0.014483   | 0.014556   |
> | 0.084000 | 0.014484   | 0.014582   |
> | 0.105000 | 0.014486   | 0.014608   |
> | 0.126000 | 0.014487   | 0.014634   |
> | 0.147000 | 0.014488   | 0.014660   |
> | 0.168000 | 0.014490   | 0.014686   |
> | 0.189000 | 0.014491   | 0.014712   |
>
> *Table: Lipschitz ratio of the model with selected layers {$0,11$} and parameter type *key* for $100$ pairs of data with $R_{\\max} = |\\theta - \\theta_0| = 0.189 $.*
>
> ---
>
> Thanks again for your time and effort. We are eager to share our insights with the community and  very much hope that you'll place us firmly in the acceptance zone.

---

> > ### Comment · Reviewer_Ptpw · 2025-08-08
> > **Thank you**
> >
> > Thank the authors for the clear and persuasive evidence and explanations. I would like to raise my score correspondingly.

---

> > > ### Author Response · Authors · 2025-08-08
> > > **Thank you**
> > >
> > > We're glad that we were able to address your requests. Thank you very much!

---

> ### Author Response · Authors · 2025-08-09
>
> We successfully replaced cross-entropy classification with MSE loss using one-hot encoding for the CoLA and SST-2 tasks, applied across different RoBERTa layers with LoRA fine-tuning. Our key finding is that MSE values below 0.2 consistently indicate strong classification performance (MSE = 0.0 for perfect predictions, MSE ≈ 0.5 for random guessing). This MSE-based framework serves as an effective substitute for cross-entropy while preserving accuracy, and it enables regression-style analysis of our theorem, which supports the inverse relationship between condition number and performance.
>
> By implementing these experiments, we have addressed your concern regarding the gap between the theoretical assumptions and the experimental setup, and we hope this will be reflected in your evaluation.
>
> | Dataset | Selected Layers | Selected Parameters | Condition Number | Train Loss | Evaluation Accuracy (%) |
> |---------|-----------------|---------------------|------------------|------------|-------------------------|
> |    Cola     | {0}             | ${W_q,W_v}$           | 11,618           | 0.18945    | 69.7987                 |
> |         | {0,11}          | $\{W_q,W_v\}$           | 9,490            | 0.18875    | 69.9904                 |
> |         | {0,5,11}        | ${W_q,W_v}$           | 7,503            | 0.18920    | 69.7987                 |
> |    | {0,5,11}        | ${W_k}$               | 2,320            | 0.18785    | 70.0863                 |
> |   SST2      | {0}             | ${W_q,W_v}$           | 5,195            | 0.16224    | 82.7982                 |
> |         | {0,11}          | ${W_q,W_v}$           | 6,413            | 0.16249    | 83.0275                 |
> |         | {0,5,11}        | ${W_q,W_v}$           | 6,792            | 0.16188    | 82.9128                 |
> |    | {0,5,11}        | ${W_k}$               | 450.64           | 0.16050    | 83.1422                 |

---

### Official Review · Reviewer_LXhB · 2025-07-02

**Clarity:** 3
**Significance:** 2
**Originality:** 3
**Rating:** 4
**Confidence:** 4

**Summary:**

The paper shows that if fine-tuning is explicitly regularized to stay near the pre-trained weights, the optimization behaves almost linearly and can be analyzed through the Neural Tangent Kernel (NTK). ,Under this linearization regime, the paper proves bounds on the empirical risk of fine-tuning as a funciton of the kernel’s eigen-spectrum. Crucially, the paper demonstrates, both theoretically and empirically, that the regularized condition number $\kappa(K + \sigma I)$ at initialization is a strong indicator for downstream performance, that could work as an effective heuristic for the selection of layers to tune.

**Questions:**

Would theorems 2-4 apply to SGD or Adam training?

What new insight does the regularization-based proximity argument provide, beyond the intuitive fact that L2 regularization encourages proximity to the pretrained model?

**Ethical Concerns:**

["NO or VERY MINOR ethics concerns only"]

**Final Justification:**

This work provides a novel kernel spectrum perspective to parameter-efficient fine-tuning. The results and insights are valuable and distinct from those offered by prior work. Therefore, this work should be published.

**Limitations:**

.

**Quality:**

3

**Strengths And Weaknesses:**

While the paper provides and interesting theoretical framework and solid proofs, the theoretical analysis contains several assumptions or logical leaps that are not sufficiently justified in the paper.
1. In section 3, the paper defines a proximal setup where the empricial risk is regularized with the L2-distance between the pretrained and fine-tuned parameters. The main argument of section 3 is that the distance between the pretrained model and the fine-tuned model is constrained by the regularization. However I see no significance of the result in this section for the following two reasons.
(i) The bounds in theorem 2~4 don't have much of a quantative meaning, because it is not for actual SGD or reasonable algorithms, but it is for an aritificial ad-hoc algorithm that is defined soley for the convinience of the proof, with no practical usage reported in prior works or through experiments in the paper. Thus the bound only has it's meaning as in theoretical demonstration for intuition.
(ii) However, the argument that adding the L2-distance of the the parameters of the pretrained model and the fine-tuned model constrains the difference between the two models is already intuitively straightforward, and thus a theoretical demonstration in a toy setup isn't really significant.
2. In sections 4 and 5, the paper implicitly equates 'performance' and the 'empirical risk'. Especially, in theorem 5, I think there should be more explanation about why the model performance should get better when the condition number is small.
Regarding the empirical results, it is indeed quite interesting that the generalization performances are predicted through the actual condition numbers of the NTK kernel, but there could have been much more the authors could have done to make a more robust and persuasive experiment. For example, for figure 2-(c), instead of comparing between different datasets, sampling 32 datasamples from the same dataset multiple times and comparing between these minidatasets could have been a more fair comparison. Also, the choice of the layers {0, 5, 11} seem quite arbitrary, it would be better to see an experiment spanning all layers of the RoBERTa-base model.


Overall, the results of the paper are interesting as it connects theoretical insight into empirical results on generalization performance and an actual methodology with practical usage, but lacks theoretical solidness and extensiveness of its experiments.


Typos
line 22: 'particularly popular methods [4, 5, 6, 7, 8] set of techniques that strive to'
line 214: 'The spectral perturbations bounds'
line 222: 'See Apendix H'
line 322: 'close to the linearization of the the pre-trained model.'
line 731: notations and definitons
line 804: 'inquality 79 will be reduced to'

---

> ### Author Rebuttal · Authors · 2025-07-31
>
> Dear Reviewer LXhB,
>
> Thank you for taking the time to review our paper. We would like to address your comments below and hope that you will consider revising your assessment upward.
>
> ---
>
> **Significance of Section 3** (i) Regarding the applicability to SGD,  while it is true that the noisy gradient aspect of stochastic approaches is not part of the theory, gradient descent/flow captures the local-search approach of gradient-based methods, including SGD and its variants. Applying selective regularization (Eq. (8)) is as straightforward as applying gradient clipping with SGD and Adam.The bounds of Theorem 3 and 4 transfer to stochastic variants through standard bounds that link their respective trajectories. (These typically need strong convexity assumptions on the loss functions and bounds on the variance of the stochastic gradient. [3]) (ii) Regarding these latter bounds being "intuitively straightforward", we would like to correct a misunderstanding. The *direct* objective of regularization is to encourage the model not to deviate but the *indirect* benefit we get is that it encourage the NTK solution and the regularized fine-tuning solutions to become closer (to answer your question, *this* is the additional insight and it is *not* straightforward.) This is what allows us to claim that the results on Section 4 apply and its direct validity is demonstrated clearly in Table 1, where we also see an additional advantage that, as long as we don't overregularize ($\\lambda$ not too large, e.g., smaller than 50), we don't sacrifice much on performance either. This advantage makes this very simple regularization very attractive for practical use (easy to implement + no sacrifice + NTK theory holds.) Note that this regularization has been proposed in an entirely different setting in reference [16] in the paper, so it is not "artificial" or "ad-hoc". We have demonstrated a fully novel way in which it can be used.
>
> ---
>
> **Condition Number vs. Performance** Theorem 5, taken as it is, does not say that the condition number dictates performance. It does say that the condition number determines the tightness of the bounds (how close the upper and lower bounds are). However, if we assume that the spectrum is mostly stable (say the arithmetic or geometric midpoint of $\\lambda_{\\textsf{min}}$ and $\\lambda_{\\textsf{max}}$ remains roughly constant), then the changes in condition number would indeed dictate the magnitude of the bounds.
>
>
> ---
>
> **Robustness to Sampling for estimating the NTK kernel** Our  work is not specifically studying optimal sketching of the kernel matrix, however, below we empirically illustrate that our  numerical results are  robust to the choice of NTK samples.  Sketching of the kernel matrices is studied in the  literature, for instance in [2]. In [1], which also modeled  fine-tuning as an NTK regression problem the number of NTK samples are fixed to $\{16,64\}$ (see Table.2 of [1]).
>
> | **Random Seed** | $\lambda_{\text{min}}$ | $\lambda_{\text{max}}$ | **Condition Number** |
> |----------------|------------------------|------------------------|-----------------------|
> | 42             | 2.04e-6                | 0.0030                 | 32.43                |
> | 123            | 6.45e-7                | 0.0035                 | 36.85                |
> | 7              | 7.37e-8                | 0.0025                 | 26.22                |
> | 99             | 3.19e-9                | 0.0024                 | 25.82                |
> | 2024           | 6.78e-8                | 0.0025                 | 26.86                |
>
> *Table: Condition numbers and eigenvalues for different random seeds $\mathbf{W}_k$ of layer 11 is fine-tuned.*
>
> ---
>
> **Extensiveness of experiments** We fine-tuned decoder-only models GPT2 and OPT-125m for 10 epochs using the Adam optimizer. LoRA with $r=8$ is used to fine-tune $\mathbf{W}_k$ of the layers $\{0,5,11\}$. The negative correlation between evaluation accuracy after 10 epochs of training and condition number of the NTK is illustrated below. In GPT2 Yelp with lowest condition number possesses the highest accuracy and in OPT-125M, IMDb and  Yelp with higher  accuracies have lower  condition  number than the other  tasks.
>
> The only exception is that IMDB in OPT-125M, despite having the smaller conditional
>
> | Dataset | Eval Accuracy (%) | Condition Number |
> |:--------|:------------------|:-----------------|
> | CoLA    | 71.0              | 83               |
> | IMDb    | 87.4              | 36               |
> | Yelp    | 87.6              | 35               |
>
> *Table: Evaluation results for GPT2.*
>
> | Dataset | Eval Accuracy (%) | Condition Number |
> |:--------|:------------------|:-----------------|
> | SST-2   | 64.8              | 910              |
> | CoLA    | 69.1              | 720              |
> | IMDb    | 70.0              | 210              |
> | Yelp    | 82.1              | 310              |
>
> *Table: Evaluation results for OPT-125m.*
>
>
>
> Thank you once again. We hope you will consider raising your score and we look forward to discussing further!
> Best,
> Authors
>
> ---
>
> [1] Malladi, S., Wettig, A., Yu, D., Chen, D. and Arora, S., 2023, July. A kernel-based view of language model fine-tuning. In International Conference on Machine Learning (pp. 23610-23641). PMLR.
>
> [2] Chen, Y., Epperly, E.N., Tropp, J.A. and Webber, R.J., 2025. Randomly pivoted Cholesky: Practical approximation of a kernel matrix with few entry evaluations. Communications on Pure and Applied Mathematics, 78(5), pp.995-1041.
>
> [3] Bottou, L., Curtis, F.E. and Nocedal, J., 2018. Optimization methods for large-scale machine learning. SIAM review, 60(2), pp.223-311.

---

> > ### Comment · Reviewer_LXhB · 2025-08-08
> >
> > Thank you for the response. The authors' rebuttal addresses my concerns and questions.
> >
> > I now see the main contribution of the results, so I have increased my score to vote to accept the paper.

---

> > > ### Author Response · Authors · 2025-08-08
> > > **Thank you**
> > >
> > > We're glad that we were able to address your questions. Thank you very much!

---

> ### Author Response · Authors · 2025-08-07
> **Awaiting your response**
>
> Dear Reviewer LXhB,
>
> We would like to thank you once again for your time and effort reading and reviewing our submission. We believe we have addressed the points that you raised. As the discussion period is ending soon, we would love to have the opportunity to clarify any remaining concerns that you might have. Could you please let us know at your earliest convenience?
>
> We look forward to sharing our work with the community and hope that you will give us a positive recommendation.
>
> Best,
> Authors

---

### Official Review · Reviewer_bqdr · 2025-07-03

**Clarity:** 3
**Significance:** 3
**Originality:** 3
**Rating:** 4
**Confidence:** 3

**Summary:**

This paper proposes a theoretical analysis of fine-tuning through the lens of linearization and NKT theory. The authors observe that by adding explicit L2 regularization to the pretrained model parameters, fine-tuning dynamics become equivalent to learning with the NTK, and establish theoretical connections between the NTK's eigenvalue spectrum and fine-tuning performance. They validate the theory empirically on LoRA fine-tuning of RoBERTa across GLUE benchmarks.

**Questions:**

Your theoretical results are derived for squared loss and gradient descent, yet your experiments succeed with cross-entropy and AdamW.  Could you provide more intuition on why your insights generalize so well to stochastic settings and different loss functions? For example, does the proposed selective regularization scheme in Equation 8 transfer straightforwardly to AdamW? What about the bounds in Thms 3 and 4?

The NTK computation uses a small sketch of only 32 samples from the training data.  How stable is the condition number to the specific choice of these 32 samples?

Table 1 shows that model accuracy is largely insensitive to the regularization strength lambda across a wide range of values. Could you explain why we expect this to be the case? Is your main takeaway here that one should always use a sufficiently large lambda that encourages linearity ?

**Ethical Concerns:**

["NO or VERY MINOR ethics concerns only"]

**Final Justification:**

I thank the authors for their detailed rebuttal and discussion. The authors should implement many additions to the current version to reflect the rebuttal and discussion. I am convinced that the reasons to accept outweigh those to reject, despite some of the weak points identified here and by other reviewers.

**Limitations:**

Yes, the authors have adequately addressed the limitations of their work. In the conclusion, they explicitly state that their theory is limited to squared loss and gradient descent, while the experiments use other configurations. They are also clear that their analysis only applies to the explicitly regularized fine-tuning they introduce, and that analyzing standard, non-regularized fine-tuning remains an open question for future work.  This is a transparent and fair assessment of the work's scope

**Paper Formatting Concerns:**

figure 2 covers part of the text on line 261

**Quality:**

3

**Strengths And Weaknesses:**

Strengths

The paper provides a theoretical contribution connecting regularized fine-tuning and NTK. The spectral perturbation bounds for layer selection are particularly interesting and may provide guidance for PEFT design. Similarly, the connection between NTK condition numbers and fine-tuning performance may offer a practical prediction tool for model selection. The experiments on RoBERTa with LoRA across multiple datasets demonstrate reasonable correspondence between theory and practice, despite some theoretical assumptions differing from experimental conditions.

Weaknesses

A primary weakness is the disconnect between the theoretical assumptions and the experimental setup. The theorems are derived for (deterministic) gradient descent with a squared loss function, while the experiments use the AdamW optimizer with a cross-entropy loss. The authors acknowledge this limitation, but it makes it unclear how tightly the theoretical bounds apply to the practical scenarios shown.
Also, the linearization analysis only applies to the explicitly regularized variant of fine-tuning introduced by the authors, and the connection to standard fine-tuning practices where such regularization may exist only implicitly is not clearly established.

The NTK condition number calculations use only 32 samples, which may not capture full kernel behavior. The paper doesn't explore how sensitive the condition number and the resulting performance predictions are to the choice of these few samples.

While the paper does a decent job of positioning itself, the distinction from previous work that also uses linearization and NTK to analyze fine-tuning (e.g., Malladi et al., https://arxiv.org/abs/2210.05643) could be sharpened. The paper's core novelty seems to be in enforcing linearity through regularization rather than just assuming it, but the practical implications of this difference could be explained more forcefully.

The experimental validation focuses on LoRA and RoBERTa-base; further investigation across different PEFT methods and model architectures would be beneficial.

---

> ### Author Rebuttal · Authors · 2025-07-31
>
> Dear Reviewer bqdr,
>
> Thank you very much for your time and effort reviewing our paper and for voicing a positive assessment. We appreciate it a lot. We hope that with what follows we address all your comments.
>
> ---
>
> **Theory vs. experiments** Our theory is designed to make a few key aspects precise: (1) that it is possible to directly elicit NTK regime behavior through regularization and (2) that, once we operate in the NTK regime, the spectrum of the kernel determines training performance. Note that (2) does not rely on any particular optimization scheme. As for (1), while it is true that the noisy gradient aspect of stochastic approaches is not part of the theory, gradient descent/flow captures the local-search approach of gradient-based methods, including SGD and its variants. Applying selective regularization (Eq. (8)) is as straightforward as applying gradient clipping with SGD and Adam. The bounds of Theorem 3 and 4 transfer to stochastic variants through standard bounds that link their respective trajectories. (These typically need strong convexity assumptions on the loss functions and bounds on the variance of the stochastic gradient. [3]) Regarding the adherence to the squared loss in both (1) and (2), note that optimizing cross-entropy is equivalent to optimizing KL-divergence, and KL-divergence and squared-loss are intimately related, given that they're both Bregman divergences. (For two outputs that are close to each other, i.e., in the high-accuracy regime, KL-divergence behaves very similarly to squared loss. [4]) As such, while the theory doesn't apply directly to cross-entropy, the behavior that we demonstrate qualitatively supports the behavior that we experimentally observe with cross-entropy.
>
> ---
>
> **Sensitivity of Performance to NTK Samples** Our  work is not specifically studying optimal sketching of the kernel matrix, however, below we empirically illustrate that our  numerical results are  robust to the choice of NTK samples.  Sketching of the kernel matrices is studied in the  literature, for instance in [2]. In [1], which also modeled  fine-tuning as an NTK regression problem the number of NTK samples are fixed to $\{16,64\}$ (see Table.2 of [1]).
>
> | **Random Seed** | $\lambda_{\text{min}}$ | $\lambda_{\text{max}}$ | **Condition Number** |
> |----------------|------------------------|------------------------|-----------------------|
> | 42             | 2.04e-6                | 0.0030                 | 32.43                |
> | 123            | 6.45e-7                | 0.0035                 | 36.85                |
> | 7              | 7.37e-8                | 0.0025                 | 26.22                |
> | 99             | 3.19e-9                | 0.0024                 | 25.82                |
> | 2024           | 6.78e-8                | 0.0025                 | 26.86                |
>
> *Table: Condition numbers and eigenvalues for different random seeds $\mathbf{W}_k$ of layer 11 is fine-tuned.*
>
>
> **Regularization and accuracy** The key takeaways from Table 1 are that, as long as we don't overregularize ($\\lambda$ not too large, e.g., smaller than 50) we simultaneously: (1) encourage the model not to deviate (the direct objective of regularization), (2) encourage the NTK solution and the regularized fine-tuning solutions to become closer (the indirect **main** objective of regularization), and (3) without sacrificing much on performance. We conjecture that the reason we don't sacrifice much on performance is that the fine-tuning solution is already relatively close to the initial model and that there are equivalently good NTK solutions in the same neighborhood. (The choice of $\\lambda$ is similar to searching for solutions within sequentially smaller neighborhoods around the initialization, as $\\lambda$ increases.)
>
> ---
>
> **Distinction from previous work** In [1] the authors assume that fine-tuning can be explained by linearization and their  results also confirm that linearization does not always appear in  fine-tuning (see fig.1 of [1]). In our work however, we prove linearization appears as result of proximity of the fine-tuned model and the pretrained model, and a regularization is  required to impose linearization. Additionally, we  study the layer selection performance by means of spectral perturbation bounds.
>
> ---
>
> The experimental validation focuses on LoRA and RoBERTa-base; further investigation across different PEFT methods and model architectures would be beneficial.
>
> **Other model architectures** We fine-tuned decoder-only models GPT2 and OPT-125m for 10 epochs using the Adam optimizer. LoRA with $r=8$ is used to fine-tune $\mathbf{W}_k$ of the layers $\{0,5,11\}$. The negative correlation between evaluation accuracy after 10 epochs of training and condition number of the NTK is illustrated below. In GPT2 Yelp with lowest condition number possesses the highest accuracy and in OPT-125M, IMDb and  Yelp with higher  accuracies have lower  condition  number than the other  tasks.
>
>
>
> | Dataset | Eval Accuracy (%) | Condition Number |
> |:--------|:------------------|:-----------------|
> | CoLA    | 71.0              | 83               |
> | IMDb    | 87.4              | 36               |
> | Yelp    | 87.6              | 35               |
>
> *Table: Evaluation results for GPT2.*
>
> | Dataset | Eval Accuracy (%) | Condition Number |
> |:--------|:------------------|:-----------------|
> | SST-2   | 64.8              | 910              |
> | CoLA    | 69.1              | 720              |
> | IMDb    | 70.0              | 210              |
> | Yelp    | 82.1              | 310              |
>
> *Table: Evaluation results for OPT-125m.*
>
> ---
>
> Thank you once again and we look forward to discussing further!
> Best,
> Authors
>
> ---
>
> [1] Malladi, S., Wettig, A., Yu, D., Chen, D. and Arora, S., 2023, July. A kernel-based view of language model fine-tuning. In International Conference on Machine Learning (pp. 23610-23641). PMLR.
>
> [2] Chen, Y., Epperly, E.N., Tropp, J.A. and Webber, R.J., 2025. Randomly pivoted Cholesky: Practical approximation of a kernel matrix with few entry evaluations. Communications on Pure and Applied Mathematics, 78(5), pp.995-1041.
>
> [3] Bottou, L., Curtis, F.E. and Nocedal, J., 2018. Optimization methods for large-scale machine learning. SIAM review, 60(2), pp.223-311.
>
> [4] Borade, S. and Zheng, L., 2008, March. Euclidean information theory. In 2008 IEEE International Zurich Seminar on Communications (pp. 14-17).

---

> > ### Comment · Reviewer_bqdr · 2025-08-07
> >
> > Thank you very much for the detailed response. I am convinced the reasons to accept outweigh those to reject, despite some of the weak points identified here and by other reviewers. I would like to maintain my positive score.

---

> > > ### Author Response · Authors · 2025-08-07
> > > **Thank you**
> > >
> > > We have addressed many of the points raised by you and other reviewers, so we would be grateful if you could reflect that in your recommendation. Regardless, we would like to sincerely thank you for engaging with us and for maintaining your positive assessment of our submission!

---

> ### Author Response · Authors · 2025-08-09
>
> We successfully replaced cross-entropy classification with MSE loss using one-hot encoding for the CoLA and SST-2 tasks, applied across different RoBERTa layers with LoRA fine-tuning. Our key finding is that MSE values below 0.2 consistently indicate strong classification performance (MSE = 0.0 for perfect predictions, MSE ≈ 0.5 for random guessing). This MSE-based framework serves as an effective substitute for cross-entropy while preserving accuracy, and it enables regression-style analysis of our theorem, which supports the inverse relationship between condition number and performance.
>
> By implementing these experiments, we have addressed your concern regarding the gap between the theoretical assumptions and the experimental setup, and we hope this will be reflected in your evaluation.
>
> | Dataset | Selected Layers | Selected Parameters | Condition Number | Train Loss | Evaluation Accuracy (%) |
> |---------|-----------------|---------------------|------------------|------------|-------------------------|
> |    Cola     | {0}             | ${W_q,W_v}$           | 11,618           | 0.18945    | 69.7987                 |
> |         | {0,11}          | $\{W_q,W_v\}$           | 9,490            | 0.18875    | 69.9904                 |
> |         | {0,5,11}        | ${W_q,W_v}$           | 7,503            | 0.18920    | 69.7987                 |
> |    | {0,5,11}        | ${W_k}$               | 2,320            | 0.18785    | 70.0863                 |
> |   SST2      | {0}             | ${W_q,W_v}$           | 5,195            | 0.16224    | 82.7982                 |
> |         | {0,11}          | ${W_q,W_v}$           | 6,413            | 0.16249    | 83.0275                 |
> |         | {0,5,11}        | ${W_q,W_v}$           | 6,792            | 0.16188    | 82.9128                 |
> |    | {0,5,11}        | ${W_k}$               | 450.64           | 0.16050    | 83.1422                 |

---

### Note · Authors · 2025-08-15

We thank all reviewers and the ACs for your work on assessing our paper and for finding our work `theoretically novel, empirically solid, and practically useful`. For your convenience, we would like to quickly summarize our contributions and the outcomes of our discussions.

---

*Novel theory* — `Connecting regularized fine-tuning to NTK` by proving that linearization can be induced and introducing `spectral perturbation bounds` for principled layer selection.

*Methodological contribution* — Easy-to-implement `selective proximity regularization` which can be empirically validated to enforce linearity as the theory predicts and, additionally, to maintain accuracy if not over-regularizing.

*Practical impact* — Revealing that the `NTK spectrum can predict downstream performance` and guide efficient model or layer selection.

*Broad experimental evidence* — Results hold across `RoBERTa, GPT-2, and OPT-125m` on GLUE and sentiment datasets.

---

*Concerns resolved* —

* Bridged `MSE–cross-entropy gap` theoretically and experimentally  — MSE-loss experiments corroborate predictions.
* Added `explicit layer-selection algorithm` as a further methodological contribution.
* Clarified `Theorems 2 and 4` with parameter trade-offs.
* Explained `condition number anomalies` — when parameter types/counts are fixed, correlation is strong to perfect.
* Demonstrated `NTK estimation robustness` to sample choice.
* Validated `Lipschitz assumptions`.

---

Thank you once more.

Best regards,

Authors

---

### Decision · Program_Chairs · 2025-09-17

**Decision:**

Accept (poster)

**Comment:**

(a) Summary

This paper proposes a theoretical analysis of Parameter-Efficient Fine-Tuning (PEFT) in large language models (LLMs) through the lens of linearization and NKT theory. The authors observe that by adding explicit L2 regularization to the pretrained model parameters, fine-tuning dynamics become equivalent to learning with the NTK, and establish theoretical connections between the NTK's eigenvalue spectrum and fine-tuning performance.  Key contributions include: (1) Theoretical bounds on the distance between fine-tuned and linearized models, (2) Characterization of fine-tuning performance via the NTK eigenvalue spectrum, and (3) Spectral perturbation bounds for layer-wise adaptation. Crucially, the paper demonstrates, both theoretically and empirically, that the regularized condition number  at initialization is a strong indicator for downstream performance, that could work as an effective heuristic for the selection of layers to tune. The paper validates the theory empirically on LoRA fine-tuning of RoBERTa across GLUE benchmarks.

(b) Strengths

The paper provides a theoretical contribution connecting regularized fine-tuning and NTK. The spectral perturbation bounds for layer selection are particularly interesting and may provide guidance for PEFT design. Similarly, the connection between NTK condition numbers and fine-tuning performance may offer a practical prediction tool for model selection. The paper rigorously formalizes PEFT dynamics through NTK linearization, deriving novel bounds for model deviation (Theorems 2–4) and risk minimization (Theorem 5). The spectral perturbation analysis (Theorems 6–7) provides actionable insights for layer selection. The experiments on RoBERTa with LoRA across multiple datasets (GLUE, IMDb, and Yelp tasks) demonstrate reasonable correspondence between theory and practice, despite some theoretical assumptions differing from experimental conditions. The results and insights are valuable and distinct from those offered by prior work.

(c) Weaknesses

Analyzing fine-tuning dynamics with NTK is not new and has been studied in e.g., [1]. And it is unclear what is the main procedure for fine-tuning with layer selection via NTK.
[1] LoRA training in the NTK regime has no spurious local minima. ICML 2024.

A primary weakness is the disconnect between the theoretical assumptions and the experimental setup. The theorems are derived for (deterministic) gradient descent with a squared loss function, while the experiments use the AdamW optimizer with a cross-entropy loss. The authors acknowledge this limitation, but it makes it unclear how tightly the theoretical bounds apply to the practical scenarios shown. Also, the linearization analysis only applies to the explicitly regularized variant of fine-tuning introduced by the authors, and the connection to standard fine-tuning practices where such regularization may exist only implicitly is not clearly established.

The NTK condition number calculations use only 32 samples, which may not capture full kernel behavior. The paper doesn't explore how sensitive the condition number and the resulting performance predictions are to the choice of these few samples.

While the paper does a decent job of positioning itself, the distinction from previous work that also uses linearization and NTK to analyze fine-tuning (e.g., Malladi et al., https://arxiv.org/abs/2210.05643) could be sharpened. The paper's core novelty seems to be in enforcing linearity through regularization rather than just assuming it, but the practical implications of this difference could be explained more forcefully.

(d) Most important reasons for your decision to accept/reject.

Theoretical rigor, with novel insights, combined with effective validation on a wide range of datasets.

(e) Summarize the discussion and changes during the rebuttal period.

Reviewer Ptpw considered that the rebuttal answered all of their concerns, and gave a high final rating of 5.
Reviewer xfCG stated that their main concerns on the theoretical results were addressed by the rebuttal, but that more (explanatory) discussions are expected in the final version.